# OracleTSC: Oracle-Informed Reward Hurdle and Uncertainty Regularization for Traffic Signal Control

**Darryl C. Jacob**                                                    *dzj0055@auburn.edu*
*Department of Computer Science and Software Engineering*
*Auburn University*

**Xinyu Liu**                                                          *xil0004@auburn.edu*
*Department of Computer Science and Software Engineering*
*Auburn University*

**Muchao Ye**                                                         *muchao-ye@uiowa.edu*
*Department of Computer Science*
*University of Iowa*

**Xiaoyong Yuan**                                                     *xiaoyon@clemson.edu*
*Department of Electrical and Computer Engineering*
*Clemson University*

**Pan He**                                                            *pan.he@auburn.edu*
*Department of Computer Science and Software Engineering*
*Auburn University*

**Reviewed on OpenReview:** *https://openreview.net/forum?id=WmJu5MkoQD*

## Abstract

Transparent decision-making is essential for traffic signal control (TSC) systems to earn public trust. However, traditional reinforcement learning–based TSC methods function as black boxes, providing little to no insight into their decisions. Although large language models (LLMs) could provide the needed interpretability through natural language reasoning, they face challenges such as limited memory and difficulty in deriving optimal policies from sparse environmental feedback. Existing TSC methods that apply reinforcement fine-tuning to LLMs face notable training instability and deliver only limited improvements over pretrained models. We attribute this instability to the long-horizon nature of TSC: feedback is sparse and delayed, most control actions yield only marginal changes in congestion metrics, and the resulting weak reward signals interact poorly with policy-gradient optimization. We introduce OracleTSC, which addresses these issues through: (1) a reward hurdle mechanism that filters weak learning signals by subtracting a calibrated threshold from environmental feedback, and (2) preventing policy degeneracy by maximizing the probability of the chosen answer, which promotes consistent decision-making across multiple responses. Experiments on the standard LibSignal benchmark demonstrate that our approach enables a compact model (LLaMA3-8B) to achieve substantial improvements in traffic flow, with a 75% reduction in travel time and 67% decrease in queue lengths over the pretrained baseline while preserving interpretability through natural language explanations. Furthermore, the method exhibits strong cross-intersection generalization: a policy trained on one intersection transfers to a structurally distinct intersection with 17% lower travel time and 39% lower queue length, all without any additional finetuning for the target topology. These findings show that uncertainty-aware reward shaping could stabilize reinforcement fine-tuning and provide a new perspective for improving its effectiveness in TSC tasks.

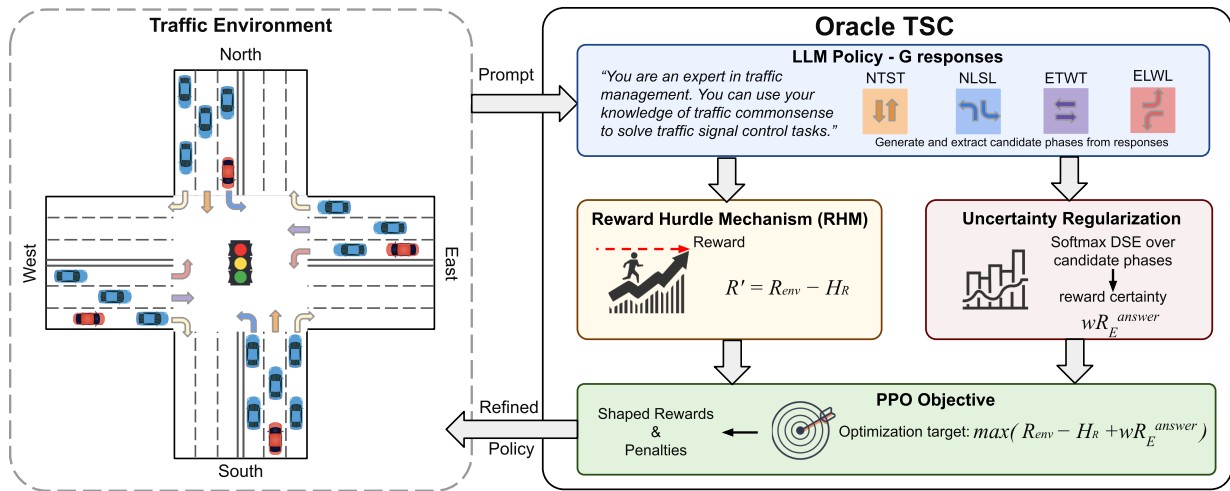

Figure 1: Overview of the OracleTSC framework. Traffic states are converted into prompts for the LLM policy, which generates candidate signal phases from $G$ sampled responses. The proposed Reward Hurdle Mechanism (RHM) emphasizes high-impact actions, while Uncertainty Regularization penalizes high-entropy responses. Both objectives are optimized jointly under the PPO framework, producing refined policies for adaptive traffic signal control.

# 1 Introduction

Most reinforcement fine-tuning on large language models (LLMs) is limited to short-horizon tasks such as question answering and summarization Cobbe et al. (2021); Hendrycks et al. (2021); Rein et al. (2024), where the outcomes of model actions are immediately observable or verifiable, thereby simplifying the credit assignment problem. In contrast to short-horizon tasks, long-horizon RL problems—such as optimizing traffic signals over extended periods Zhang et al. (2019); Mei et al. (2024); He et al. (2024)—pose unique challenges. Although recent pioneering studies have explored applying LLMs to traffic signal control (TSC) Wang et al. (2024a); Lai et al. (2025b), these methods often rely on external components—such as pretrained critic models for policy learning and trajectory filtering, or ensembles with conventional TSC algorithms for assisted decision-making. Such dependencies limit the TSC system's autonomy and scalability and arise from practical constraints: LLMs applied to TSC must make decisions whose long-term utility becomes evident only after many timesteps. However, eliciting such long-horizon behavior using gold-standard Chain-of-Thought (CoT) Wei et al. (2022b) annotations is impractical, as the state–action space expands exponentially with simulation length.

**Key Observations and Insights.** While Proximal Policy Optimization (PPO) is a natural framework for long-horizon TSC, we find that it struggles to produce consistent gains when optimizing LLM-based policies under realistic traffic dynamics, consistent with recent observations in LLM-driven TSC Lai et al. (2025b). Our empirical analysis further identifies two recurring failure modes that are especially pronounced.

*(1) Weak reward signals reinforce suboptimal actions.* In TSC, most feasible phase changes lead to only small improvements—or slight deteriorations—in congestion measures such as average queue length. As a result, reward signals are weak and often masked by randomness, producing very small policy-gradient updates. Instead of strengthening the few actions that truly help, PPO tends to spread learning across many actions with little impact. This problem is made worse by delayed traffic responses and strong temporal dependence: the effect of a signal change may appear only many steps later, making it difficult to assign credit correctly. Consequently, learning progresses slowly and may even stall, despite extended training.

*(2) Reasoning drift under output uncertainty.* We further observe that when the LLM policy is uncertain, its generated explanations can differ widely across responses for the same traffic state, resulting in inconsistent

phase choices. We refer to this behavior as reasoning drift. It injects randomness directly into the action selection process and allows errors to accumulate over time. Notably, producing longer or more detailed reasoning does not resolve this issue. Stable decisions emerge only when the model's output becomes focused on a single action. In practice, this appears as high variability across responses during early training and is reduced only when uncertainty is explicitly penalized.

Together, these observations highlight challenges that are not typically encountered in short-horizon or rule-based reinforcement learning tasks, but are central in TSC systems where rewards are sparse, delayed, and weakly informative, and where action consistency across time is critical for stable TSC control.

To address these limitations, we introduce **OracleTSC**, a traffic signal control framework that uses a reward hurdle to filter out low-impact updates and a Softmax Discrete Semantic Entropy regularizer to stabilize reasoning. Together, these components produce more robust and consistent control policies with clearer, more reliable explanations across varied intersection settings. We evaluate **OracleTSC** on LIBSIGNAL Mei et al. (2024), the standard benchmark for TSC. Results show that adjusting rewards for time-varying inflows, under a hurdle rate, combined with minimizing semantic entropy — measured as the presence of multiple distinct answer modes under stochastic token generation, improves both control performance and the consistency of the model's explanations.

**Contributions**. To summarize, our contributions are as follows:

- While prior studies have highlighted the challenges of finetuning PPO for long-horizon, TSC tasks with time-varying traffic patterns, we identify insufficient suppression of suboptimal actions as a key factor underlying this difficulty. To address this issue, we introduce a simple penalty that imposes an explicit performance threshold, penalizing actions falling below a baseline. By suppressing low-quality policy updates, our introduced Reward Hurdle Mechanism (RHM) amplifies the learning signal from high-impact actions, leading to substantial performance gains.

- We propose a principled approach to quantifying and mitigating uncertainty across responses by introducing the semantic entropy reward via temperature-scaled softmax as a regularization term in policy optimization. This *uncertainty-aware mechanism* curbs drift in long-horizon reasoning and enhances TSC performance across multiple metrics, including shorter queue lengths and reduced travel times.

- We show that our OracleTSC not only enhances policy stability but also delivers consistent performance gains across diverse model scales, effectively shortening queue lengths and improving the consistency of generated explanations. The model also demonstrates strong cross-intersection generalization, performing well when trained on one intersection and tested on a structurally distinct new intersection.

## 2 Related Work

**Traffic Signal Control.** Early TSC systems relied on rule-based heuristics and fixed-time schedules that were unable to adapt to time-varying traffic patterns Board et al. (2015); Martinez et al. (2011). Classical algorithms such as Max Pressure (MP) control Varaiya (2013) improved throughput by activating signal phases proportional to weighted queue differences, yet remained limited to local optimization.

Deep reinforcement learning (DRL) introduced adaptive decision-making based on learning from experience. IntelliLight Wei et al. (2018) pioneered this direction by using real-world video-derived data to train deep Q-networks, while PressLight Wei et al. (2019a) extended MP theory into its reward function to guide efficient signal control. Subsequent work improved generalization and coordination: AdLight Wang et al. (2023) leveraged movement-level augmentation to encode fine-grained vehicle interactions, and Wang et al. (2024b); Jiang et al. (2024) proposed junction-matrix and topology-mapping methods that enabled zero-shot transfer across heterogeneous intersections. Other advances emphasized robustness and sample efficiency—FuzzyLight Li et al. (2025a) used fuzzy logic to handle noisy sensors, while DreamerV3 Li et al. (2025b) incorporated world models to perform latent rollouts, reducing environment simulation costs. Importantly, our objective is not to claim superiority over all existing RL algorithms or fully optimized black-box

RL systems. Highly optimized, non-explainable RL controllers remain strong on several benchmarks. Rather, our goal is to improve the performance, stability, and reliability of explainable LLM-based controllers while preserving natural-language reasoning and semantic-level uncertainty modeling.

Recent work has begun exploring the use of LLMs to overcome these limitations through explicit reasoning and explainable decision-making. LLMLight Lai et al. (2025a) demonstrated that prompting pretrained LLMs with structured traffic states enables natural-language explanations of phase selection, while VLMLight Wang et al. (2025) extended this to multimodal, vision-language reasoning for safety-critical intersections. Yet these approaches rely primarily on zero-shot prompting or supervised imitation, lacking the closed-loop optimization needed for long-horizon policy improvement. Weak or noisy reward signals slow convergence, allowing suboptimal reasoning to overwhelm beneficial actions in long-horizon tasks. Standard PPO stabilizers, such as clipped ratios, reward normalization, and advantage scaling, alleviate but do not eliminate training drift, leaving LLM-based TSC vulnerable to unstable gradients under weak and delayed rewards. This exposes a shared challenge between control variance and epistemic uncertainty. Motivated by this connection, we treat uncertainty not merely as a diagnostic signal but as a direct optimization objective. While PPO and reward shaping are well established in reinforcement learning, our contribution lies in showing that targeted reward thresholding and uncertainty regularization can substantially improve the stability of LLM-based traffic signal control.

Beyond decision-making and control, recent work has explored using LLMs for automatic algorithm design and heuristic discovery. A recent systematic survey Liu et al. (2026) highlights how LLMs can generate executable algorithms, code-based solutions, and structured reasoning processes that improve both interpretability and responsiveness. Similarly, *Evolution of Heuristics* Liu et al. (2024) demonstrates that LLMs can iteratively refine algorithmic strategies, enabling efficient automatic design of problem-specific solvers.

These approaches focus primarily on synthesizing or evolving discrete algorithmic procedures. In contrast, traffic signal control requires stable policy optimization under weak, delayed reward feedback in a stochastic control environment. Our work is complementary: rather than designing new symbolic algorithms, we study how to directly stabilize reinforcement fine-tuning of LLM-based controllers, enabling autonomous, interpretable decision-making without reliance on external critics or hybrid ensembles.

**Uncertainty Quantification and Reduction**. A parallel line of research focuses on improving reasoning reliability and trustworthiness in LLMs through uncertainty quantification and entropy minimization. Unsupervised fine-tuning frameworks such as EMPO Zhang et al. (2025) and RENT Prabhudesai et al. (2025) remove the need for labeled reward models by minimizing entropy at the answer and token levels, respectively, enabling confidence-driven optimization. Building on CoT prompting Wei et al. (2022a), UnCert-CoT Zhu et al. (2025) leverages token-level entropy signals to trigger additional reasoning steps, while *Uncertainty of Thoughts* Hu et al. (2024) models the reasoning process as a decision tree in which the LLM generates self-queries and selects the branch that maximizes entropy reduction in the final answer. Semantic entropy Kuhn et al. (2023); Farquhar et al. (2024) further refines these principles by exploiting bidirectional entailment between reasoning paths and answer candidates as a proxy for the true entropy over semantic classes. This approach links semantic entropy values to the probability of model error, providing a theoretically grounded measure of confidence. Meanwhile, Kernel Language Entropy (KLE) Nikitin et al. (2024) generalizes this concept by computing von Neumann entropy over a kernelized Laplacian of a semantic graph, where edge weights represent the degree of entailment between responses. Collectively, these approaches reveal a strong coupling between accuracy and confidence Ye et al. (2025), motivating entropy reduction as a core regularization objective in reasoning-focused training.

Across both domains, *OracleTSC* attempts to unify advances in TSC and uncertainty-aware reasoning. Its RHM subtracts a fixed threshold from rewards to filter weak updates, reducing gradient variance and stabilizing long-horizon PPO optimization. Complementing this, the *Softmax-with-Temperature Discrete Semantic Entropy* regularizer extends entropy-minimization methods such as EMPO Zhang et al. (2025) and RENT Prabhudesai et al. (2025) to sequential control by adding an uncertainty-based bonus to an environmental reward. This coupling of reward filtering and entropy-based feedback grounds semantic consistency in RL, yielding interpretable, confidence-calibrated policies that improve stability and long-horizon performance beyond prior LLM-based TSC systems.

## 3   Method

To address TSC with LLMs, we begin with a key observation: although general-purpose LLMs demonstrate strong natural language reasoning capabilities, they lack the precision and domain awareness required for TSC decision-making. To bridge this gap, we develop a domain-specific reinforcement fine-tuning framework that enables LLMs to acquire optimal phase-selection policies through direct interaction with simulated traffic environments. We follow a variant of PPO Schulman et al. (2017), modified to preserve the LLM's instruction-following ability. Central to our approach is the idea that minimizing uncertainty and suppressing subpar actions enhances both reasoning quality and policy performance.

### 3.1   Task Formulation

Given a TSC environment, we represent traffic states, reasoning, and control actions as natural language text and formulate LLM-based TSC at timestep $t$ as a Markov Decision Process (MDP) $(\mathcal{S}, \mathcal{A}, P, r, \gamma)$. The state space $\mathcal{S} \subseteq [1, V]^L$ consists of token sequences representing traffic intersection states, where $V$ is the vocabulary size and $L$ is the maximum sequence length. The action space $\mathcal{A} \subseteq [1, V]$ contains the next candidate token generated by the policy.

For a maximum response length $O$, the token-level state evolves as

$$s_{l+1}^t = \begin{cases} s_0^t, & l = 0, \\ \text{concat}(s_l^t, a_l^t), & 1 \leq l < O, \end{cases}$$

where $l$ indexes a generated output token. Here, $s_0^t$ denotes the initial token sequence formed by concatenating the system prompt with the tokenized traffic state, and $\text{concat}(\cdot)$ denotes token concatenation subject to the length constraint $O$. The system prompt specifies the task instructions, output format for phase selection, and the current as well as the two most recent traffic states and corresponding actions (see Listing 3). The traffic state reports the number of vehicles in each lane. Following Lai et al. (2025b), each lane is subdivided into four portions based on vehicle position and motion. The first portion consists of early queued vehicles, defined as vehicles traveling below 0.1 m/s (i.e., waiting at the intersection). The second portion includes vehicles within 10% of the road length from the signal. The third portion contains vehicles located between 10% and 33% of the road length from the signal. The fourth portion comprises vehicles that are more than 33% of the road length away from the signal.

The procedure for extracting the selected phase from the output tokens is outlined in Algorithm 2. This formulation naturally induces the state transition function $P : \mathcal{S} \times \mathcal{A} \to \mathcal{S}$. The reward function $r : \mathcal{S} \times \mathcal{A} \to \mathbb{R}$ evaluates the agent's performance at each token. At timestep $t$, the agent observes $s_l^t \in \mathcal{S}$, selects $a_l^t \in \mathcal{A}$ via policy $\pi : \mathcal{S} \to \mathcal{P}(\mathcal{A})$, and receives

$$r(s_l^t, a_l^t, l) = \begin{cases} -\beta\, \text{KL}\big(\pi_\theta(a_l^t \mid s_l^t) \,\big\|\, \pi_{\text{REF}}(a_l^t \mid s_l^t)\big), & 0 \leq l < O, \\ R(s_O^t, a_O^t), & l = O, \end{cases} \tag{1}$$

where $\pi_{\text{REF}}$ is a frozen reference model and $\beta$ controls the KL penalty. Thus, rewards combine (i) intermediate KL regularization to stabilize generation and (ii) a final task-specific signal $R(s_O^t, a_O^t)$ measuring phase effectiveness (e.g., congestion reduction). The objective is to learn $\pi^*$ that maximizes the expected discounted return $G_l = \mathbb{E}\Big[\sum_{l=1}^O \gamma^l r(s_l^t, a_l^t, l)\Big]$, with discount $\gamma \in [0, 1]$.

We aim to train a LLM-based agent $\pi_\theta : \mathcal{S} \to \mathcal{A}$ that generates the selected traffic phase and a corresponding CoT explanation. A traffic phase is defined as "a controller timing unit associated with the control of one or more movements" USF, where a movement refers to a specific permitted vehicle direction (e.g., eastbound through, northbound left-turn). The policy is parameterized by Low-Rank Adapter (LoRA) Hu et al. (2022) weights $\theta$, which are injected into all linear layers of the backbone LLM (e.g., the pre-trained Qwen3-0.6B Yang et al. (2025)). Detailed descriptions of the prompt templates and sample inputs are provided in the Appendix.

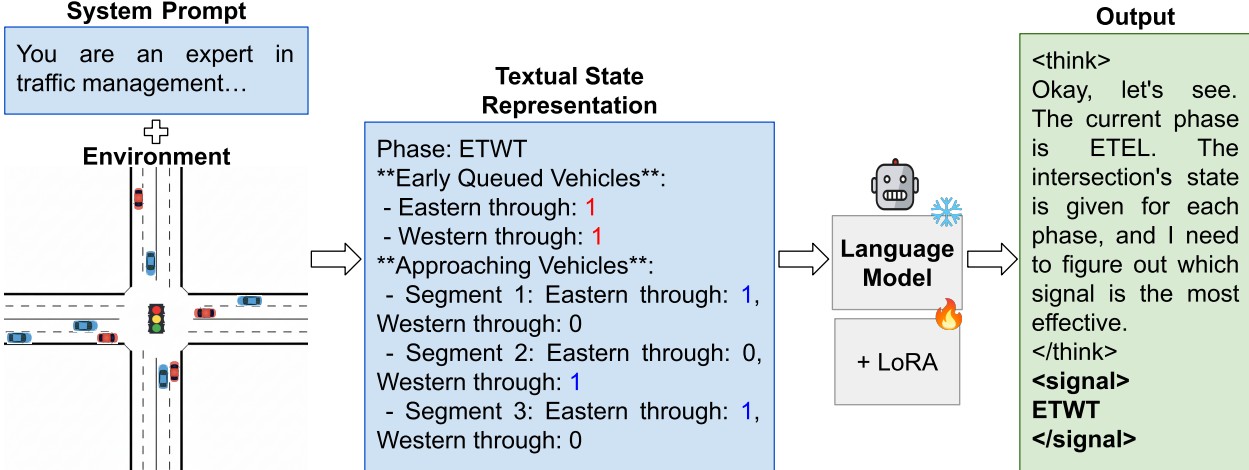

Figure 2: The environment state is translated into a structured textual traffic representation and provided to the LLM, including lane-wise early queued vehicles and segmented approaching vehicles. The model then generates a chain-of-thought reasoning trace, analyzing queue imbalances, anticipating near-term inflow from upstream segments, and evaluating candidate signal phases before producing a final phase selection.

## 3.2 Training Objective

We train our LLM-based agent to allocate its token budget toward effective reasoning that minimizes queue lengths at intersections. Following PPO, we optimize the policy network $\pi_\theta$ while jointly training a value network $V_\phi$ to reduce variance and improve stability.

Training proceeds at the level of individual state–response exchanges. Given a traffic state $s_t$, the LLM generates a single response trajectory $\kappa = \{(s_0, a_0), \ldots, (s_O, a_O)\}$ consisting of reasoning tokens followed by a phase selection. Each exchange is treated as an independent training instance, similar in spirit to contextual bandit-style reinforcement fine-tuning used in RLHF. We do not propagate gradients across multiple environment steps or perform trajectory-level temporal credit assignment over extended horizons.

For notational simplicity, we omit the timestep superscript $t$ from $s_{l+1}^t$ in the following discussion. Specifically, given a response trajectory $\kappa = \{(s_0, a_0), \ldots, (s_O, a_O)\}$ rolled out under $\pi_{\text{old}}$ (the current LoRA-adapted LLM), the clipped surrogate objective for the policy is

$$\mathcal{J}_{\text{CLIP}}(\theta) = \hat{\mathbb{E}}_{\kappa \sim \pi_{\text{old}}} \left[ \min\big(r_l(\theta)\,\hat{A}_l,\ \text{clip}(r_l(\theta),\, 1 - \epsilon,\, 1 + \epsilon)\,\hat{A}_l\big) \right], \tag{2}$$

where

$$r_l(\theta) = \frac{\pi_\theta(a_l \mid s_l)}{\pi_{\text{old}}(a_l \mid s_l)}, \qquad \epsilon = 0.2,$$

and $\hat{\mathbb{E}}$ denotes the empirical expectation over minibatches of tokens sampled from trajectories. The clipping term constrains the policy update to remain within the trust region defined by $(1 - \epsilon, 1 + \epsilon)$. When the likelihood ratio $r_l(\theta)$ moves outside this interval, the objective is flattened so that excessively large policy shifts do not receive disproportionately large gradients. This prevents unstable oscillations in the log-probabilities of selected actions and helps maintain monotonic policy improvement, consistent with the motivations of Trust Region Policy Optimization Schulman et al. (2015a) and PPO Schulman et al. (2017). We estimate advantages using Generalized Advantage Estimation (GAE) Schulman et al. (2015b), where temporal differences are defined as

$$\delta_l = r_l + \gamma V_\phi(s_{l+1}) - V_\phi(s_l) \tag{3}$$

with $V_\phi(s_{O+1}) = 0$, and advantages are computed as

$$\hat{A}_l = \sum_{k=0}^{O-l} (\gamma\lambda)^k \delta_{l+k} \tag{4}$$

with discount $\gamma \in [0, 1]$ and GAE weight $\lambda \in [0, 1]$.

We employ Generalized Advantage Estimation (GAE) to stabilize token-level policy updates within a single response trajectory. Although traffic control is a long-horizon problem at deployment time, our optimization operates at the exchange level: advantages are computed over the generated reasoning sequence for the current decision only. We do not accumulate discounted returns across multiple environment transitions.

Within a response trajectory, Monte Carlo token-level returns often suffer from high variance, whereas fully bootstrapped value targets can introduce substantial bias. Generalized Advantage Estimation (GAE) provides a principled trade-off between these extremes through the parameter $\lambda$, improving the stability of token-level policy updates without introducing cross-exchange temporal dependencies.

Conversely, relying solely on bootstrapped value predictions introduces bias, as the value network $V_\phi$ is trained to approximate returns under a previous policy rather than the updated one. GAE provides a principled mechanism for balancing this bias–variance trade-off through the parameter $\lambda$, interpolating between low-variance, high-bias temporal-difference estimates ($\lambda = 0$) and high-variance, low-bias Monte Carlo estimates ($\lambda = 1$).

This bias–variance trade-off is important in LLM-based control because reasoning sequences are long in token space, even though optimization is performed at the level of independent exchanges. The bootstrapped return target is then $\hat{G}_l = \hat{A}_l + V_\phi(s_l)$.

We train the value network by minimizing a clipped regression loss:

$$\mathcal{J}_{\text{value}}(\phi) = \tfrac{1}{2}\,\hat{\mathbb{E}}\Big[\max\Big(\big(V_\phi(s_l) - \hat{G}_l\big)^2,\ \big(\text{clip}(V_\phi(s_l) - \hat{G}_l, -\epsilon_v, \epsilon_v)\big)^2\Big)\Big], \tag{5}$$

with a separate clipping hyperparameter $\epsilon_v$ (e.g., $\epsilon_v = 0.2$).

**Overall objective.** Our overall objective is to minimize directly

$$\mathcal{L}(\theta, \phi) = -\,\mathcal{J}_{\text{CLIP}}(\theta) + \alpha\,\mathcal{J}_{\text{value}}(\phi), \tag{6}$$

with $\alpha \geq 0$.

Importantly, optimization is performed independently for each exchange; future traffic states influence learning only through the scalar reward assigned to the current decision, rather than through multi-step temporal backpropagation across environment transitions.

Although entropy bonuses are standard in PPO for discrete action spaces, extending them to LLM-based policies is nontrivial because actions correspond to multi-token response trajectories rather than single categorical choices. Prior work, such as Discrete Semantic Entropy (DSE) Zhang et al. (2025), adapts entropy regularization to LLM outputs by aggregating uncertainty over semantically distinct answers. However, these formulations are often too coarse to reliably penalize low-quality reasoning trajectories and do not explicitly model the smoothness or concentration of the induced answer distribution. As a result, they provide limited control over response-level instability in long-horizon decision-making. Motivated by these limitations, we introduce two complementary mechanisms in the following sections that directly target weak learning signals and reasoning instability in LLM-based TSC.

### 3.3 Reward Hurdle

PPO for LLMs Luong et al. (2024) is most commonly applied to short-horizon settings (e.g., context-conditioned question answering). In long-horizon control, however, naive application of PPO is insufficient: early suboptimal actions can push the system into regimes where later improvements are severely limited.

In TSC, once queues approach or exceed traffic capacity, even principled controllers such as MaxPressure Varaiya (2013) struggle to recover; the robust strategy is to keep queues low proactively. This aligns with our preliminary experiments, where PPO often rewarded trajectories showing only minor *reductions* in queue length (small positive $R(s_O^t, a_O^t)$). The optimization thus failed to produce policies that preemptively cleared queues, instead yielding idle strategies that led to traffic saturation, where subsequent actions could no longer alleviate congestion.

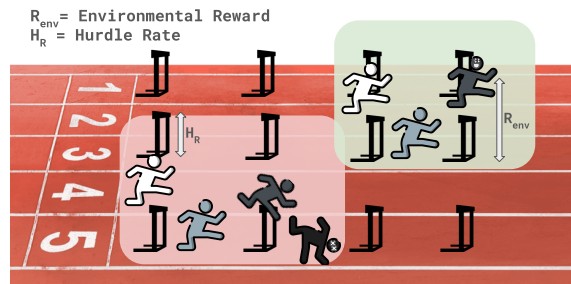

To address this, we introduce the Hurdle Rate $H_R$ as a constant that shifts the sequence-level reward, ensuring that only trajectories surpassing a minimum improvement threshold receive positive reinforcement, while suboptimal ones are penalized relative to baseline. This encourages the LLM to explore more aggressive, high-impact actions. Specifically, the original environmental reward used in PPO at timestep $t$ is defined as

$$R_{\text{env}}(s_O^t, a_O^t) = \text{queue}_{t-1} - \text{queue}_t. \tag{7}$$

Figure 3: Each runner represents a trajectory; a hurdle represents the improvement threshold $H_R$. Runners that consistently clear hurdles ($R_{\text{env}}(s_O^t, a_O^t) \geq H_R$) advance, illustrating why the hurdle mechanism shifts the policy toward higher-impact actions. Contacting a hurdle (failing to exceed $H_R$) incurs a penalty and reduces progress, analogous to a sub-hurdle sequence receiving negative reinforcement that propagates backward through tokens.

We define $\text{queue}_t$ as the average number of vehicles whose speed falls below 0.1 m/s across all lanes at timestep $t$. We use the *queue difference* between timesteps $t$ and $t-1$ to mitigate instabilities in the reward signal induced by time-varying traffic demand drift. Without this adjustment, increasing inflows may introduce spurious penalties that incorrectly discourage otherwise reasonable actions. We incorporate a hurdle rate $H_R$ and yield the reward defined as:

$$R(s_O^t, a_O^t) = R_{\text{env}}(s_O^t, a_O^t) - H_R. \tag{8}$$

Accounting for time-varying inflow-induced regime shifts alone does not substantially reduce congestion. As shown in Figure 4(a) for the Qwen3–0.6B model on the CityFlow1×1 benchmark, nearly 70% of actions reduce queues by fewer than 2.5 vehicles—far too weak to produce meaningful policy updates. The remaining 30% of actions exceed this 2.5-vehicle threshold, but they occur too infrequently to drive reliable long-horizon improvement. To amplify the learning signal associated with these higher-impact actions, we use 2.5 vehicles as an initial reference point and introduce a constant $H_R$: actions that fail to surpass this threshold within each control interval are down-weighted.

As shown in Figure 4(b), subtracting this hurdle substantially sharpens the reward distribution. The proportion of actions exceeding the 2.5-vehicle threshold increases from 30% to over 40% after a single training episode—indicating a clear shift toward higher-impact interventions. In practice, we determine $H_R$ through a combination of empirical tuning and reward-distribution analysis, using the 70% statistic only as an initial guide rather than a fixed rule. This procedure provides a principled cutoff for distinguishing impactful from low-impact actions while adapting across environment–model pairs.

**Connection to Variance Reduction Techniques.** Following Greensmith et al. (2004), an effective baseline for policy gradient methods should (i) depend only on the state rather than the sampled actions, and (ii) be selected to minimize the variance of the policy gradient estimator for $\mathcal{J}_{\text{CLIP}}$:

$$\text{Var}[\nabla_\theta \mathcal{J}_{\text{CLIP}(\theta)}] \approx \text{E}_{\tau \sim \pi_\theta} \Big[ \sum_{l=1}^{O} \nabla_\theta \log \pi_\theta(a_l^\tau | s_l^\tau) \Big( \sum_{k=l}^{O} r(s_k^\tau, a_k^\tau) - b(s_k^\tau) \Big) \Big] \tag{9}$$

A common choice in PPO is to use a state-dependent learned value function $V_\phi(s)$ as the baseline, i.e., an estimator of the expected future return. However, as noted in Greensmith et al. (2004), other forms of baselines can also effectively reduce gradient variance. In particular, Greensmith et al. (2004) derives an

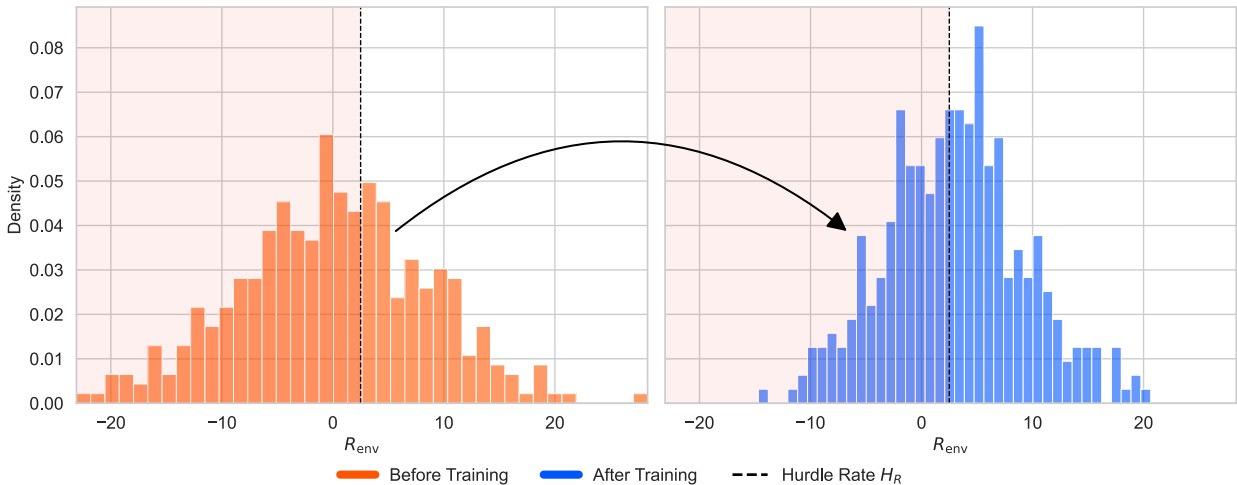

Figure 4: Empirical distribution of queue length differences with Hurdle $H_R = 2.5$ from a baseline episode on CityFlow1x1 using pretrained Qwen3-0.6B and a testing episode on CityFlow1x1 after finetuning Qwen3-0.6B for one episode using the $R_{\text{env}}(s_O^t, a_O^t) - H_R$ configuration. Since decisions are taken every 10 timesteps, 360 decisions are taken over $T = 3600$ timesteps in an episode.

*optimal constant baseline*—a single scalar value (per policy) that minimizes the variance of the policy-gradient estimator. Their result, presented in Theorem 13 for the GPOMDP gradient $\nabla \pi_\theta / \pi_\theta$, can be rewritten in the equivalent $\nabla \log \pi_\theta$ form via a standard transformation:

$$b(s_k^\tau) = \frac{\mathrm{E}_{\tau \sim \pi_\theta, \tau_0 = s_k^\tau}\left[ \sum_{l=1}^{O} ||\nabla_\theta \log \pi_\theta(a_l^\tau | s_l^\tau)||_2^2 \left( \sum_{k=l}^{O} r(s_k^\tau, a_k^\tau) \right) \right]}{\mathrm{E}_{\tau \sim \pi_\theta, \tau_0 = s_k^\tau}\left[ \sum_{l=1}^{O} ||\nabla_\theta \log \pi_\theta(a_l^\tau | s_l^\tau)||_2^2 \right]} \tag{10}$$

Although the expression includes expectations over trajectories, the baseline itself is a *single constant value* for a fixed policy, not a function of $s$. Greensmith et al. (2004) refer to it as a "constant baseline" because, despite being defined through trajectory-level expectations, it does not vary across states. While theoretically appealing, computing this optimal constant baseline is computationally infeasible in practice. To overcome this limitation, we blend the benefits of both state-dependent and constant forms. Early in training—when $V_\phi$ is still inaccurate—we stabilize updates by subtracting a fixed *Hurdle Rate $H_R$*, which functions as a constant baseline. As training progresses and $V_\phi$ improves, the value function **naturally** begins to dominate the baseline term in the advantage estimate, providing a state-dependent adjustment that better captures long-horizon structure. Thus the transition between the constant and state-dependent baselines arises automatically: the Hurdle Rate provides stability when the value function is unreliable, and $V_\phi$ increasingly guides the updates once it becomes accurate. Conceptually, the Hurdle Rate plays a role similar to that of the optimal constant baseline, with one key distinction: instead of being subtracted from every reward in the trajectory, it is applied only to the final environmental reward of each trajectory $\kappa$.

### 3.4 Semantic Entropy Reward via Temperature-Scaled Softmax

A major challenge in long-horizon traffic control is that standard entropy-based regularizers measure uncertainty at the token level rather than at the level of the actual decision. Token uncertainty often reflects linguistic variation rather than uncertainty about the traffic-phase choice. As a result, token entropy poorly estimates semantic confidence and can destabilize training, especially when the same phase is expressed with different wording across responses. To address this limitation, we measure uncertainty at the level of the semantic action (Figure 5)—the traffic phase predicted by the model—rather than at the level of the text token. Our approach relies on a simple assumption: if the model predicts the same traffic phase across multiple responses, it is semantically confident; if it alternates between phases, it is not.

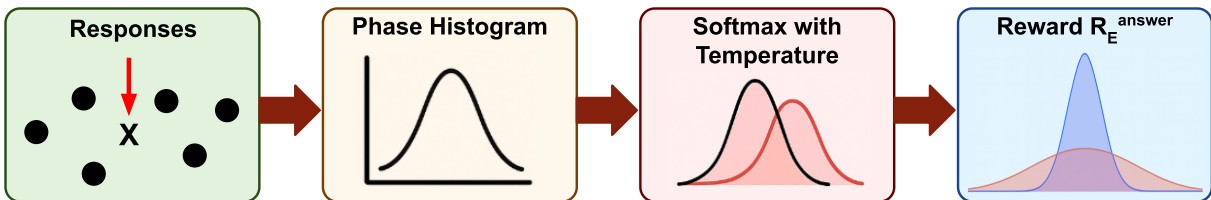

Figure 5: Uncertainty Regularization. Responses are first separated by extracted phase to estimate the probability of the LLM responding with a particular action. We quantify the spread of their distributions using the Softmax with Temperature Discrete Semantic Entropy. We apply a reward term $R_E^{\text{answer}}$ to encourage confident policies during optimization.

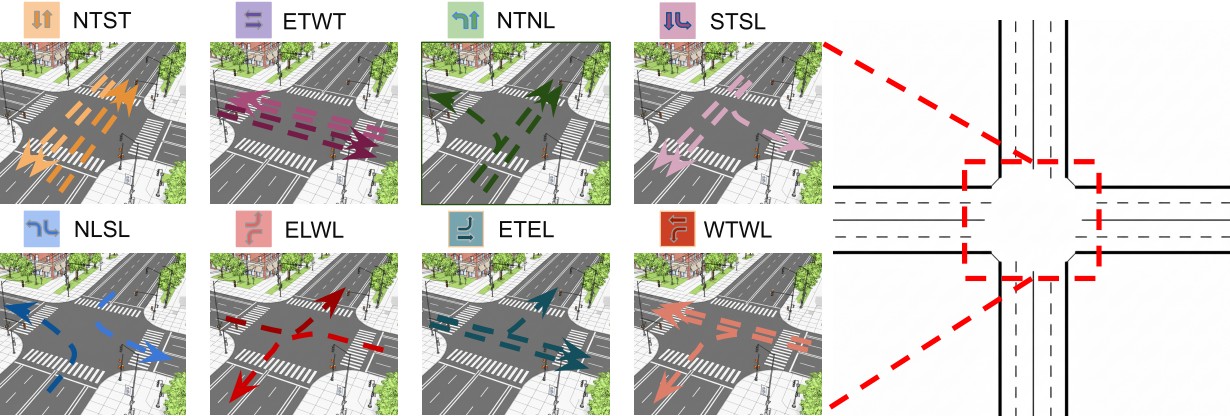

Figure 6: Visualization of eight supported signal phases at the intersection in the CityFlow 1×1 benchmark.

Formally, given two output responses, $e_1^t = s_0 \cup \{a_0, \ldots, a_O\}$ and $e_2^t = s_0 \cup \{a'_0, \ldots, a'_O\}$, featuring the same tokenized traffic intersection state $s_0$, we can cluster both responses based on the phase we extract from the answer $e$. This step is essential because the same phase can be expressed in many linguistic forms; without grouping responses by their underlying phase, we cannot tell whether the model is semantically consistent or simply paraphrasing. Clustering converts free-form responses into a discrete distribution over phase choices, allowing us to compute a meaningful measure of semantic entropy. A list of supported phases for the CityFlow 1x1 intersection are provided in Figure 6. Finally, not all uncertainty should be treated equally. A distribution such as $(0.55, 0.45)$ may be acceptable in some contexts but signal high uncertainty in others. To control sensitivity to disagreement across responses, we apply temperature-scaled normalization to the phase counts, producing a tunable, decision-level entropy measure that can be directly incorporated into the reward.

In our study, we sample eight output token sequences $e_1^t, e_2^t, \ldots, e_8^t$ and extract a single phase from each. These extracted phases are treated as integers from 1 to $N_P$ (inclusive) to serve as discrete cluster labels. We estimate $[p_1, \ldots, p_j, \ldots, p_{N_P}]$ using the empirical probability of each phase when extracted from each of the eight output responses given the same state:

$$\text{count}_j^{\text{answer}} = \sum_{g=1}^{G=8} \mathbb{I}(\text{phase}_g == j)$$

$$p_j^{\text{answer}} = \frac{\exp\left(\text{count}_j^{\text{answer}}/\tau\right)}{\sum_{j'=1}^{N_P} \exp\left(\text{count}_{j'}^{\text{answer}}/\tau\right)} \tag{11}$$

where $\mathbb{I}$ denotes the indicator function that is 1 when we extract the $j$-th phase from the $g$-th output response and 0 otherwise, and $\tau$ is a temperature hyperparameter that controls how much we prioritize marginal changes in the empirical count of an extracted phase.

Given phase probabilities for the DSE over extracted phases to proxy the LLM's uncertainty over which phase to report. Since we use the phase extracted from the first response as the chosen TSC action, we mark the chosen phase as $j_c$. We turn these probabilities into a reward as follows:

$$R_E^{\text{answer}}(e_1^t, \ldots, e_G^t, R_{env}, H_R) = \begin{cases} p_{j_c}^{\text{answer}} & R_{env} > H_R \\ 0 & \text{otherwise} \end{cases} \qquad (12)$$

The answer entropy reward uses the probability of the selected phase $p_{j_c}^{\text{answer}}$ only when the environmental reward $R_{env}$ (measured as the reduction in queue length) exceeds the hurdle threshold $H_R$. Otherwise, the reward is set to zero. This mechanism prevents reinforcing poorly performing actions that are generated with high confidence, effectively decoupling prediction certainty from control quality.

**Discussion and Relation to DSE and Self-Questioning.** The classic DSE Zhang et al. (2025) serves as a natural baseline, computing entropy directly from empirical action counts, which is defined as

$$R_E^{\text{answer,DSE}} = \frac{\text{count}^{\text{answer}} j}{\sum_{j'=1}^{N_P} \text{count}_{j'}^{\text{answer}}} \qquad (13)$$

While effective in unsupervised question-answering settings, DSE relies on manually tuned entropy clipping and is sensitive to dataset-specific entropy scales. We generalize this idea by applying a temperature-scaled Softmax to the phase frequencies, yielding a more flexible confidence measure. The temperature $\tau$ creates a continuum between two behaviors. At moderate temperatures, the reward mirrors DSE, weighting phases by frequency. As $\tau \to 0$, the Softmax approaches an $\arg\max$, producing a self-questioning–style Chen et al. (2025) reward that activates only when the chosen phase matches the majority vote. As $\tau \to \infty$, the distribution becomes uniform and insensitive to agreement. This unified formulation connects semantic-entropy minimization Zhang et al. (2025) with self-consistency maximization Chen et al. (2025), providing a single, tunable mechanism that encourages both confidence and internal consensus—two properties essential for stabilizing long-horizon control in traffic signal optimization.

After incorporating the two mechanisms, our final total reward is then formulated as

$$R(s_O^t, a_O^t) = R_{\text{env}}(s_O^t, a_O^t) - H_R + w_E R_E^{\text{answer}}, \qquad (14)$$

where $w_E$ is a hyperparameter that balances the impact of uncertainty.

## 4 Experiments

We empirically investigate three key questions: **(Q1)** Does minimizing *Temperature-Scaled Softmax Discrete Semantic Entropy* improve traffic-signal control performance compared to training without an uncertainty regularizer? **(Q2)** Does introducing a reward hurdle on the environmental reward improve performance by suppressing low-impact trajectories? **(Q3)** Does our approach improve the consistency and informativeness of the LLM's CoT rationales (e.g., cross-response agreement and alignment between reasoning and actions)?

### 4.1 Experimental Setup

#### 4.1.1 Testbeds

We conduct experiments using LibSignal Mei et al. (2024), a unified benchmark for traffic-signal reinforcement learning that includes widely used algorithms such as IDQN Vincent et al. (2023), CoLight Wei et al. (2019b), PressLight Wei et al. (2019a), and MaxPressure Varaiya (2013). LibSignal provides an extensible Gym-style observation–action interface, allowing us to integrate natural-language policies by adding a custom answer-extraction mechanism (Algorithm 2). We evaluate our method on two contrasting intersections: **CityFlow1 ×1**: a compact, four-approach intersection in Hangzhou, China, characterized by symmetric inflow patterns and an eight-phase signal plan controlling through and left-turn movements (Figure 7); **Cologne1**: a more irregular and structurally complex intersection in Cologne, Germany, featuring mixed through, right-turn, left-turn, and U-turn movements across four approaches (Figure 7). LibSignal incorporates real-world traffic data from each corresponding region—Hangzhou for CityFlow 1×1 and Cologne for

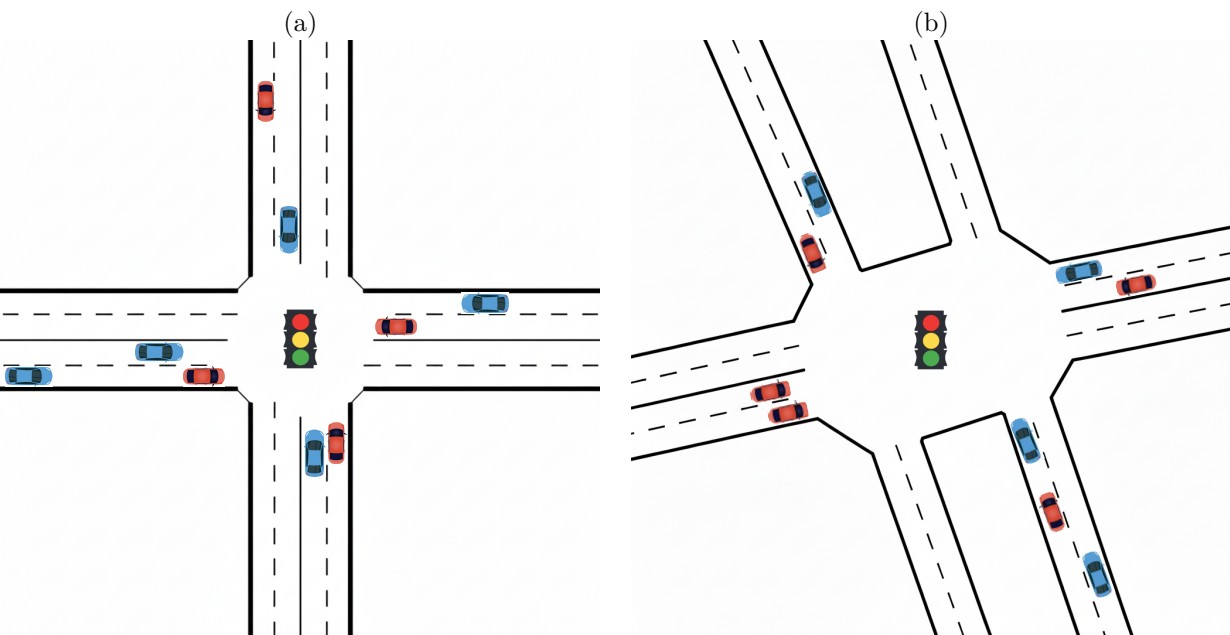

Figure 7: Visualization of the road network at a selected intersection in the (a) CityFlow 1×1 and (b) Cologne 1 Benchmark.

Cologne1—generating realistic inflow patterns and driver behaviors and providing a challenging testbed for long-horizon control.

### 4.1.2 Evaluation Metrics

We evaluate performance using four standard traffic-signal metrics following LibSignal Mei et al. (2024). **Travel time** is the average duration required for a vehicle to complete its assigned route. **Queue length** is computed as the average number of vehicles whose speed falls below $0.1\,\text{m/s}$ across all lanes and timesteps. **Delay** follows the U.S. Federal Highway Administration (FHWA) definition as the additional travel time experienced relative to free-flow conditions USF; that is, the average difference between actual travel time and its free-flow counterpart. **Throughput** denotes the total number of vehicles that successfully complete their routes during the simulation.

### 4.1.3 Implementation Details

We fine-tune LoRA adapters Hu et al. (2022) with rank 16 on top of pretrained LLM backbones rather than updating all model parameters. All experiments are conducted on a single NVIDIA RTX A6000 GPU. Each traffic simulation runs for $3,600$ timesteps, and the policy selects an action every 10 steps. When a phase change is issued, a fixed 5-second yellow interval is inserted to ensure safe transitions. Policy and value networks are optimized using the objective in Equation 6. Rewards are computed according to Equation 14 with $\alpha = 1$ and discount factor $\gamma = 0.999$. We perform parameter updates every 360 timesteps using a replay buffer containing the most recent 400 timesteps of state–action–reward tuples, and we save LoRA and value-head checkpoints every 720 timesteps. The selected phase is extracted directly from the LLM's output using Algorithm 2, without additional model calls.

### 4.1.4 TSC Baselines

We evaluate three backbone LLMs—Qwen3-0.6B, Qwen3-8B, and LLaMA3-8B-Instruct—both before and after reinforcement-learning fine-tuning. All models use identical prompts, reward functions, and training protocols. During inference, we follow the decoding hyperparameters recommended by each model's authors

Table 1: Performance of TSC agents in CityFlow and Cologne.

| Method | CityFlow 1×1 | | | | Cologne1 | | | |
|---|---|---|---|---|---|---|---|---|
| | Travel Time↓ | Queue↓ | Delay↓ | Throughput↑ | Travel Time↓ | Queue↓ | Delay↓ | Throughput↑ |
| Fixed-control Transportation Methods | | | | | | | | |
| FixedTime (t_fixed=10) | 923.67 | 114.42 | 4.83 | 1006 | 206.46 | 51.59 | 4.04 | 1847 |
| FixedTime (t_fixed=30) | 552.72 | 100.34 | 5.19 | 1455 | 156.96 | 46.23 | 4.22 | 1910 |
| MaxPressure | 303.81 | 100.94 | 6.77 | 1717 | 58.34 | 7.69 | 2.77 | 2002 |
| Non-explainable RL Methods | | | | | | | | |
| SOTL | 212.22 | 69.11 | 5.42 | 1861 | 1358.62 | 104.61 | 5.94 | 515 |
| IDQN | 116.64 | 28.37 | 0.61 | 1959 | 50.58 | 5.37 | 0.31 | 2000 |
| MAPG | 490.71 | 118.61 | 0.74 | 1514 | 46.73 | 5.06 | 0.30 | 2000 |
| IPPO | 308.87 | 95.38 | 0.76 | 1736 | 55.74 | 7.35 | 0.33 | 1994 |
| PressLight | 105.68 | 23.20 | 0.58 | 1965 | 51.79 | 4.82 | 0.29 | 2001 |
| FRAP | 104.28 | 23.29 | 0.62 | 1979 | 66.67 | 12.09 | 0.31 | 1995 |
| Explainable LLM Methods | | | | | | | | |
| Qwen3-0.6B | 506.1 | 104.06 | 0.71 | 1485 | 531.8 | 40.61 | 0.57 | 1548 |
| Qwen3-8B | 468.95 | 98.91 | 0.72 | 1566 | 119.4 | 34.37 | 0.43 | 1919 |
| LLaMA3-8B | 594.9 | 138.63 | 0.81 | 1359 | 74 | 13.52 | 0.36 | 1995 |
| Hurdle and Softmax Probability (Qwen3-0.6B) | 204.1 (−60%) | 70.4 (−32%) | 0.73 (+3%) | 1897 (+28%) | 80.7 (−85%) | 20.89 (−49%) | 0.42 (−26%) | 1990 (+29%) |
| Hurdle and Softmax Probability (Qwen3-8B) | 187.9 (−60%) | 66.26 (−33%) | 0.76 (+6%) | 1902 (+21%) | 84.3 (−29%) | 21.25 (−38%) | 0.42 (−2%) | 1987 (+4%) |
| Hurdle and Softmax Probability (LLaMA3-8B) | 146.9 (−75%) | 45.50 (−67%) | 0.67 (−17%) | 1946 (+43%) | 52.9 (−29%) | 6.73 (−50%) | 0.32 (−11%) | 2001 (+<1%) |

(e.g., temperature, top-$k$, top-$p$), ensuring that downstream performance differences arise from policy optimization rather than decoding artifacts. For context, we compare our RL-finetuned models against widely used non-LLM traffic-signal controllers, including IDQN, PressLight, FRAP, and other fixed-time or rule-based baselines. Unlike these non-explainable controllers, our fine-tuned LLMs produce both high-quality control actions and interpretable chain-of-thought rationales.

## 4.2 Main Results

We train policies on two different road network topologies: CityFlow1×1 and Cologne1. LibSignal provides real-world traffic data from an intersection in Hangzhou with $N_P = 8$ phases, and from an intersection in Cologne with $N_P = 4$. Our method trains policies using the objective in Equation 6 with $\alpha = 1$, where each reward $r_t$ incorporates both the environmental reward and the hurdle rate $H_R$. For computing Temperature-scaled Softmax DSE, we set $G = 8$ as a balance between computational cost and estimation quality: larger values of $G$ reduce the variance of the estimated output phase distribution but increase wall-clock time and memory usage. To avoid undersampling control modes, we follow the heuristic $G \geq N_P$, where $N_P$ is the number of admissible traffic phases ($N_P$=8 for CityFlow1×1). We apply the same setting to Cologne1 for consistency. Each traffic simulation episode lasts $3,600$ timesteps, and the agent selects an action every 10 timesteps. When the LLM proposes a phase change, a yellow-light period of five seconds is inserted to transition safely from the current phase. We report agent performance using the best checkpoint, selected based on the lowest average queue length during a held-out test episode.

Table 1 shows that OracleTSC consistently improves all major metrics across models and intersections. On CityFlow1×1, applying both the Reward Hurdle and Temperature-Scaled Softmax DSE yields a $60 - 75\%$ reduction in travel time, a $32 - 67\%$ reduction in queue length, and substantial gains in throughput. On Cologne1, which is structurally different and has fewer phases, the method again reduces queue length by $29 - -50\%$ and improves travel time despite the more constrained action space. Importantly, the gains persist across model scales: even the 0.6B-parameter Qwen model benefits significantly, demonstrating that the proposed mechanisms operate independently of model size.

## 4.3 Ablation Studies

We utilize ablation studies to study the impact of our design choices on the effectiveness of our method. We conduct our ablation studies on the CityFlow1×1 configuration with LLaMA3-8B as the base LLM.

Table 2: Ablation study over reward design. $R_E^{\text{answer}}$ is given by Softmax DSE with $\tau = 0$. Base LLM is LLaMA3-8B. Evaluated on CityFlow1×1.

| Total Reward $R$ | Travel Time | Queue Length | Delay | Throughput |
|---|---|---|---|---|
| Baseline | 594.9 | 138.63 | 0.81 | 1359 |
| $R_{\text{env}}(s_O^t, a_O^t)$ | 391.1 | 117 | 0.77 | 1611 |
| $R_{\text{env}}(s_O^t, a_O^t) - H_R(= 2)$ | 174.8 | 56.9 | 0.7 | 1901 |
| $R_{\text{env}}(s_O^t, a_O^t) + w_{\text{uncertainty}} R_E^{\text{answer}}$ | 204 | 71.1 | 0.68 | 1823 |
| $R_{\text{env}}(s_O^t, a_O^t) - H_R(= 3) + w_{\text{uncertainty}} R_E^{\text{answer}}$ | **146.9** | **45.5** | **0.67** | **1946** |

### 4.3.1 Reward Shaping

Table 2 summarizes an ablation over the key reward components in OracleTSC. We compare: (i) the pre-trained LLM without RL finetuning, (ii) finetuning using only the environmental reward, (iii) adding a constant hurdle rate, (iv) applying Temperature-scaled Softmax DSE as an uncertainty regularizer, and (v) their combination.

Using the environmental reward alone yields only limited performance gains. Although average travel time decreases from 594.9 s to 391.1 s, congestion-related metrics remain: queue length is reduced only marginally ($138.63 \rightarrow 117$), and delay exhibits negligible change ($0.81 \rightarrow 0.77$). These results indicate that, in long-horizon TSC, the raw environmental reward is both weak and noisy, and therefore insufficient to reliably guide policy optimization.

Introducing a hurdle rate fundamentally alters the optimization landscape. By subtracting $H_R$, low-impact actions are explicitly penalized, preventing the policy from reinforcing transitions that induce only marginal improvements. As a result, learning is redirected toward decisions that produce meaningful congestion reduction. This leads to substantial gains across all metrics, most notably a reduction in queue length from 138.63 to 56.9, confirming that reward shaping is critical for mitigating PPO's tendency to overfit to weak positive signals in long-horizon settings.

Uncertainty regularization alone, implemented via temperature-scaled Softmax DSE, also improves performance relative to the baseline. Queue length decreases from 138.63 to 71.1, suggesting that a significant source of PPO instability with LLM-based policies stems from reasoning drift and inconsistent action generation. Penalizing high-entropy response distributions encourages more stable and coherent phase selection, even in the absence of explicit reward shaping.

The strongest performance is achieved when the hurdle rate and uncertainty regularization are combined. This joint objective yields consistent improvements across all metrics: travel time is reduced from 594.9 s to 146.9 s, and queue length drops from 138.63 to 45.5 vehicles. These results demonstrate that the two mechanisms are complementary: the hurdle rate strengthens informative learning signals, while uncertainty regularization stabilizes the LLM's internal reasoning process and reduces gradient variance.

Overall, these results indicate that entropy regularization alone delivers consistent performance improvements, and that Temperature-scaled Softmax DSE provides a stronger and more stable training signal than naïve DSE. More broadly, the findings underscore the importance of managing epistemic uncertainty in LLM-based control policies—particularly in long-horizon domains where PPO is sensitive to variance in both generated reasoning and trajectory-level rewards.

### 4.3.2 Comparison to naive Discrete Semantic Entropy

We further analyze the role of entropy regularization by comparing the proposed temperature-scaled Softmax DSE against the previously introduced naive DSE. All methods are evaluated under a unified reward formulation $R = R_{\text{env}} + w_{\text{uncertainty}} R_E^{\text{answer}}$, with $w_{\text{uncertainty}} = 1$. Experiments are conducted using LLaMA3-8B-Instruct on the CityFlow-$1 \times 1$ benchmark, and results are summarized in Figure 8.

Across all episodes, entropy-based regularization consistently outperforms the pretrained baseline, yielding faster convergence and improved traffic efficiency. In Episode 1, naive DSE reduces average travel time

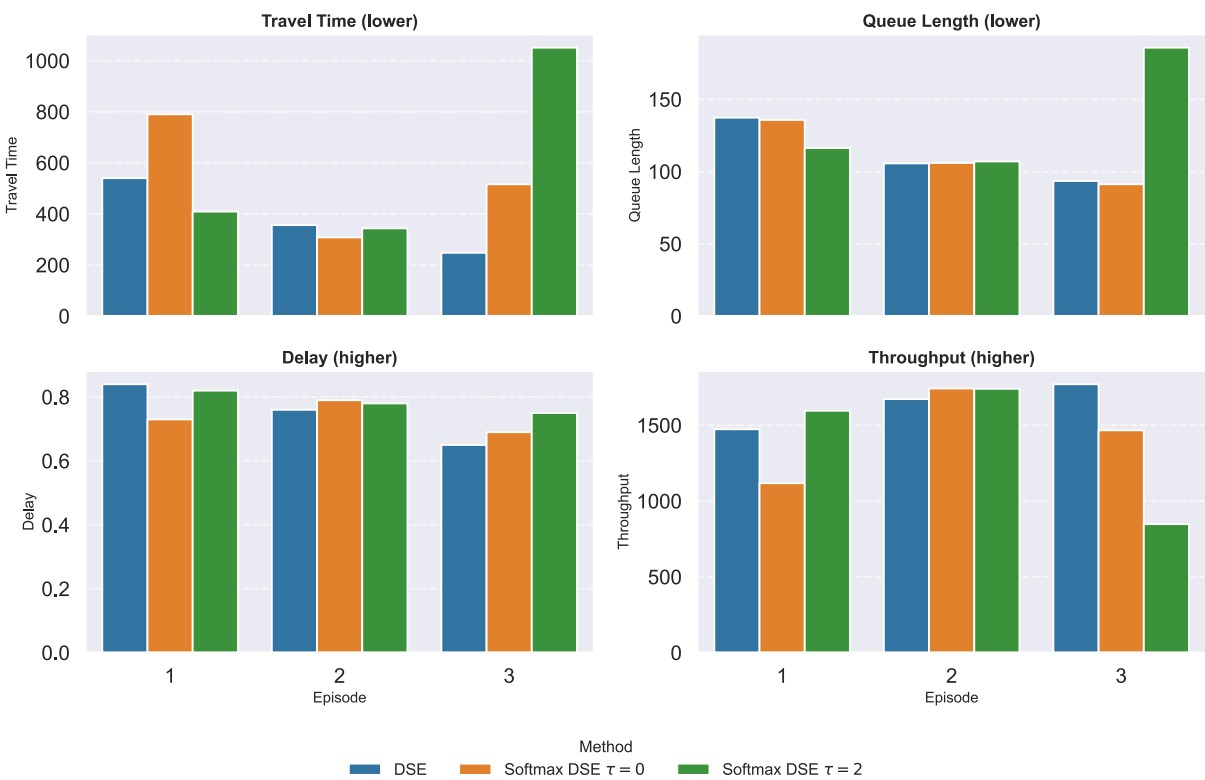

Figure 8: Effect of entropy regularization and temperature scaling across training episodes under the $R_{\text{env}}(s_O^t, a_O^t) + w_{\text{uncertainty}} R_E^{\text{answer}}$ configuration where $w_{\text{uncertainty}} = 1$ and $H_R = 0$. Each subplot reports one evaluation metric (Travel Time, Queue Length, Delay, and Throughput) as a function of the hurdle rate for LLaMA3-8B on the CityFlow1×1 intersection. The results indicate that entropy regularization alone is effective, while the sharpest probability distribution over phases ($\tau = 0$) yields the strongest overall performance in terms of Queue Length.

from 594.9 s to 540.4 s, while the temperature-scaled Softmax DSE ($\tau = 2$) further lowers it to 409.4 s and increases throughput from 1,359 to 1,596 vehicles. By Episode 2, both uncertainty-aware methods continue improving performance, with naive DSE achieving 356.5 s travel time and 1,673 throughput, while the $\tau = 2$ variant further improves to 343.8 s and 1,741 vehicles. In Episode 3, uncertainty minimization sustains these gains, reducing travel time to 248.0 s and reaching a peak throughput of 1,771 vehicles.

Notably, the sharpest response distribution ($\tau = 0$) attains the lowest queue length in Episode 3 (91.29 vehicles), despite slightly worse travel time and throughput, suggesting that overly aggressive entropy suppression can favor local consistency over beneficial exploration. This limitation is mitigated by incorporating the hurdle rate, which filters low-impact trajectories and stabilizes optimization under low-entropy regimes.

Overall, these results confirm that entropy regularization is a key driver of performance improvement, and that temperature-scaled Softmax DSE provides a more effective and stable learning signal than naive DSE. This highlights the critical role of uncertainty control in LLM-based policies, particularly for long-horizon decision-making problems where PPO is highly sensitive to output variance and compounded reward noise.

### 4.3.3 Hurdle Rate

We next investigate the influence of the hurdle rate $H_R$ under the reward formulation $R = R_{\text{env}} - H_R + w_{\text{uncertainty}} R_E^{\text{answer}}$, with $w_{\text{uncertainty}} = 1$. Figure 9 summarizes results across a wide range of hurdle values for three backbone models: Qwen3-0.6B, Qwen3-8B, and LLaMA3-8B.

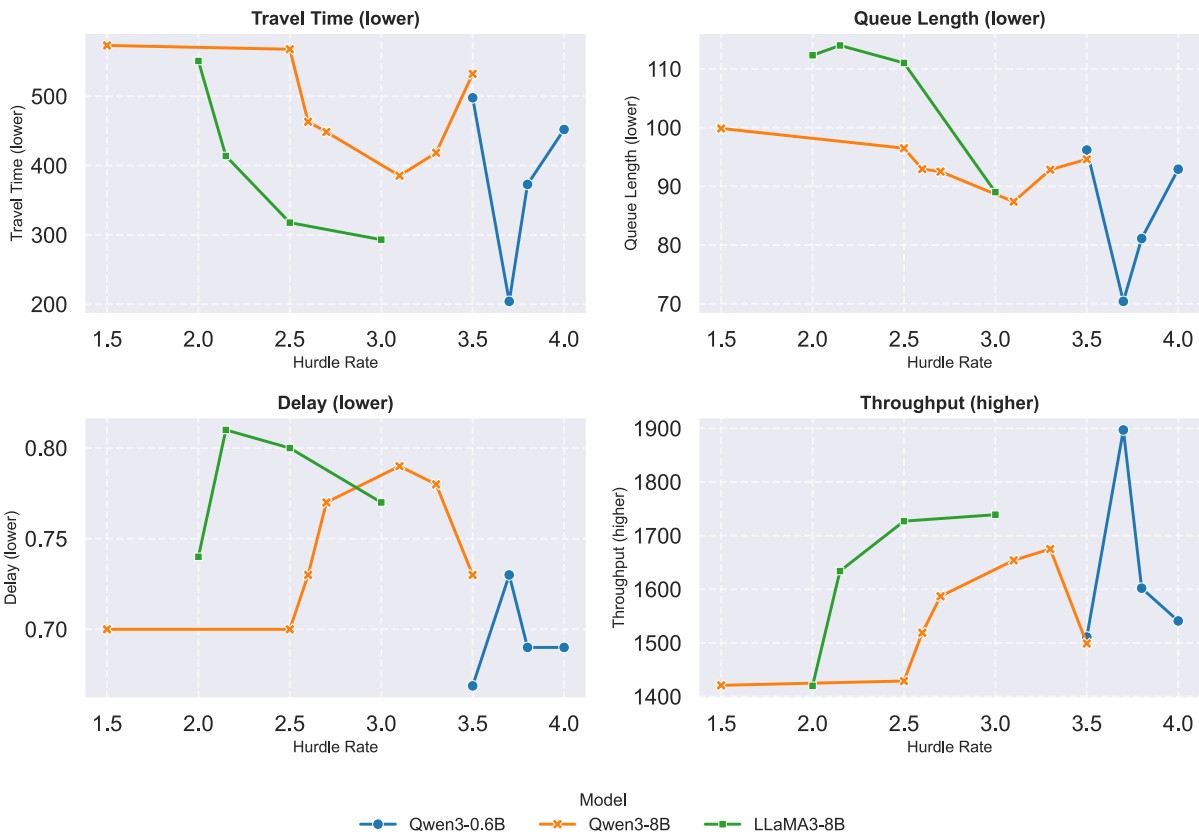

Figure 9: Effect of the Hurdle rate on model performance for the $R_{\text{env}}(s_O^t, a_O^t) - H_R + w_{\text{uncertainty}} R_E^{\text{answer}}$ configuration where $w_{\text{uncertainty}} = 1$. Each subplot reports one evaluation metric (Travel Time, Queue Length, Delay, and Throughput) as a function of the hurdle rate for Qwen3-0.6B, Qwen3-8B, and LLaMA3-8B on the CityFlow1×1 intersection. The results indicate that moderate hurdle rates consistently balance exploration and reward exploitation, leading to shorter travel times and reduced congestion compared with both low and high thresholds, though optimal performance requires careful tuning.

Across all architectures, a consistent trend emerges: intermediate hurdle rates yield the strongest performance. This behavior aligns with variance-reduction theory in policy gradients Greensmith et al. (2004), where the variance of the gradient estimator increases with the squared deviation between the optimal constant baseline $b^*$ and a mis-specified baseline $b$. In our setting, the hurdle rate plays a role analogous to a baseline, shaping the effective learning signal.

For Qwen3-0.6B, performance peaks around $H_R \approx 3.7$, reducing average travel time from 497.8 s to 204.1 s and queue length from 96.2 to 70.4 relative to the lowest tested hurdle ($H_R = 3.5$), while increasing throughput from 1,511 to 1,897. Qwen3-8B achieves optimal performance at slightly lower thresholds ($H_R \approx 2.7$), reaching a travel time of 385.4 s and throughput of 1,675. LLaMA3-8B exhibits similar behavior, attaining its best results at $H_R = 3.0$, with a travel time of 293.1 s in a representative training episode.

These results reveal two complementary failure modes associated with miscalibrated hurdle rates. When $H_R$ is too low, weak or marginal trajectories are insufficiently penalized, allowing noisy rewards to dominate PPO updates and slow convergence. In contrast, overly aggressive hurdle rates suppress informative transitions, impairing coordination and leading to increased travel times and reduced throughput across all models.

Model scale further modulates the optimal hurdle threshold. Smaller models, such as Qwen3-0.6B, benefit from slightly higher hurdle rates, whereas larger models attain peak performance at more moderate values. We attribute this divergence to differences in intrinsic reasoning stability: larger models typically produce

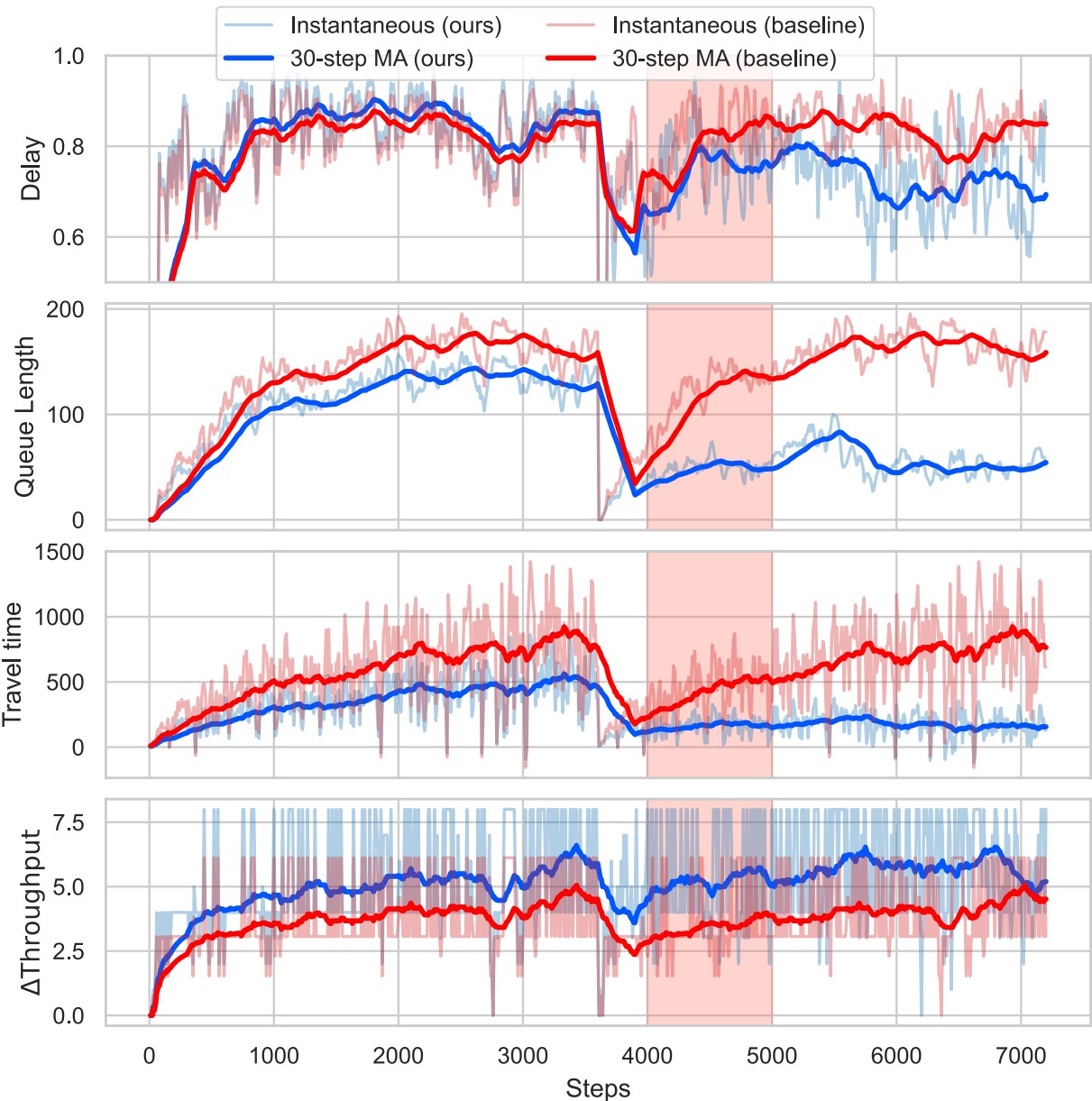

Figure 10: Training dynamics of the LLM-based traffic signal control agent over two full episodes. Each subplot reports instantaneous metrics for the finetuned LLM (in blue) and a 30-step moving average for (a) delay, (b) queue length, (c) travel time, and (d) throughput change. We also include the baseline LLM's dynamics (in red) The shaded region highlights the onset of a large improvement in traffic efficiency, marked by reduced delay, shorter queues, and larger and more consistent throughput gains. Together, these curves illustrate how the agent transitions from unstable exploratory behavior to stable, reward-aligned control.

more coherent and internally consistent action sequences prior to reinforcement fine-tuning, and are therefore more susceptible to performance degradation when informative trajectories are excessively pruned.

Taken together, these findings demonstrate that the hurdle rate is not a fixed hyperparameter, but rather a model- and topology-dependent control mechanism that actively shapes the policy optimization landscape.

Proper calibration is crucial for balancing exploratory learning against the suppression of low-value updates, thereby enabling stable and efficient optimization in long-horizon, sparse-reward traffic signal control.

## 4.4 Comparison to Other RL Controllers

Table 1 compares our LLM-based traffic signal controllers against fixed-time baselines and non-LLM reinforcement learning methods across two intersection settings. On CityFlow-$1\times1$, fixed-time controllers exhibit poor performance, characterized by long average travel times ($552-924$ s) and severe queueing. Classical deep-RL approaches, including IDQN, PressLight, and FRAP, substantially outperform fixed-time control, achieving travel times of approximately $100-120$ s and maintaining short queues of roughly $23--28$ vehicles. In contrast, pretrained LLM backbones are initially uncompetitive: for example, LLaMA3-8B yields an average travel time of 594.9 s with a queue length of 138.63 vehicles, demonstrating that naïve prompting alone is insufficient for effective TSC. After applying the proposed reward-hurdle mechanism and entropy-regularized fine-tuning, however, the same backbone improves markedly, reaching 146.9 s average travel time, a queue length of 45.5 vehicles, and a throughput of $1,946$ vehicles. This closes most of the performance gap relative to specialized deep-RL controllers, while uniquely preserving interpretable natural-language rationales for each control decision.

On the more irregular Cologne1 intersection, the gap to strong RL baselines narrows further. PressLight and MaxPressure obtain travel times of 51.79 s and 58.34 s, respectively, with near-maximal throughput ($\approx 2,000$ vehicles). Our fine-tuned LLaMA3-8B controller achieves performance on par with this regime, attaining an average travel time of 52.9 s, a queue length of 6.73 vehicles, and a throughput of 2,001 vehicles, while substantially outperforming its pretrained counterpart (74 s, 13.52 vehicles, and 1,995 vehicles, respectively). Smaller backbones exhibit similar trends: for example, Qwen3-0.6B improves from 531.8 s to 80.7 s travel time and from 1,548 to 1,990 vehicles in throughput on Cologne1. These results show that (i) untrained LLM policies fall far short of established RL controllers, but (ii) after RL fine-tuning with our reward hurdle and uncertainty regularizer, OracleTSC attains performance comparable to strong non-LLM RL baselines on realistic networks, while simultaneously providing interpretable chain-of-thought explanations.

Figure 10 illustrates how these gains emerge over time as the agent transitions from unstable exploration to stable, reward-aligned control.

## 4.5 Generalization across Intersections

Table 3: Generalization performance across intersections for LLaMA3-8B under the Environmental Reward - $H_R$ + Softmax DSE configuration. Each row reports evaluation metrics when a policy trained on one traffic network is directly applied to another without additional finetuning. Models trained on Cologne1 generalize well to CityFlow$1\times1$, improving travel time from 594.9 s to 517.5 s and throughput from 1359 to 1427. Conversely, transferring from CityFlow$1\times1$ to Cologne1 yields further gains in efficiency (travel time 61.3 s vs. 74 s and queue length 8.27 vs. 13.52), suggesting that the learned representations capture transferable signal-phase coordination patterns across distinct network topologies.

| Testing Intersection | Variant | Travel Time | Queue Length | Delay | Throughput |
|---|---|---|---|---|---|
| CityFlow$1\times1$ | Baseline | 594.9 | 138.63 | 0.81 | 1359 |
| | Trained on Cologne1 | **517.5** | **137.25** | **0.8** | **1427** |
| Cologne1 | Baseline | 74 | 13.52 | 0.36 | 1995 |
| | Trained on CityFlow$1\times1$ | **61.3** | **8.27** | **0.34** | **2002** |

We evaluate the cross-topology generalization capabilities of OracleTSC by training policies on one intersection and deploying them on another without any additional finetuning. Table 3 reports zero-shot transfer results for LLaMA3-8B under the $R_{\text{env}} - H_R + w_{\text{uncertainty}} R_E^{\text{answer}}$ configuration.

A policy trained on Cologne1 transfers effectively to CityFlow$1\times1$, improving travel time from 594.9 s to 517.5 s and increasing throughput from $1,359$ to $1,427$. This demonstrates that representations learned on a structurally heterogeneous intersection can generalize to a simpler, more symmetric geometry with measurable gains in efficiency. The reverse transfer (CityFlow$1\times1 \rightarrow$ Cologne1) yields even stronger improvements:

travel time decreases from 74 s to 61.3 s, queue length is reduced by nearly 40% (13.52 $\rightarrow$ 8.27 vehicles), and throughput increases from $1,995$ to $2,002$—the highest throughput achieved among all RL and non-RL controllers (explainable or not)

These bidirectional improvements suggest that combining hurdle-based shaping with uncertainty minimization leads the policy to acquire transferable phase-selection behaviors rather than memorizing intersection-specific patterns. The ability to deploy a model zero-shot across geometrically distinct networks indicates that OracleTSC learns stable decision-making heuristics that generalize beyond the training topology. These findings provide early evidence for the feasibility of scalable, reusable LLM-based traffic-signal controllers that require minimal per-intersection adaptation.

### 4.6 Qualitative Results

We qualitatively analyze how incorporating environmental rewards, a hurdle rate, and Softmax DSE regularization refines the model's internal reasoning and decision-making trajectory.

Before fine-tuning (Listing 1), the model's reasoning trace exhibits self-contradictions, vacillation, and redundant phase re-evaluation. It frequently reverses intermediate conclusions (e.g., "**Wait...**"), misidentifies phase mnemonics (e.g., **ETL, WTL** should be **ETEL** and **WTWL**), and repeatedly re-analyzes the same input block (**Phase:ETWT**), inefficiently consuming output tokens and demonstrating weak adherence to the prompt. Collectively, these behaviors reflect diffuse credit assignment and elevated epistemic uncertainty in the model's chain-of-thought.

After fine-tuning (Listing 2), the CoT becomes markedly structured, grounded, and concise. The model explicitly enumerates reasoning steps (e.g., "Step 1: Identify... Step 2: The optimal traffic signal..."), anchors each inference in observable quantities ("(segment 3) is 4, and there are no early queued vehicles"), and concludes decisively with a single aligned action, <signal>ETEL</signal>. The absence of reversals or reanalysis suggests sharper credit propagation under the hurdle objective and reduced output entropy from the Softmax DSE term.

The generated chain-of-thought is not guaranteed to faithfully reflect the model's internal reasoning process, and ensuring mechanistic faithfulness in LLM reasoning remains an open research challenge. Recent surveys highlight the difficulty of scaling inference, supervising reasoning processes, and building reliable agentic systems Ke et al. (2025). Similarly, studies of the extractive–abstractive spectrum reveal inherent trade-offs among abstraction, verifiability, and perceived utility in LLM generations Worledge et al. (2024). Accordingly, we interpret OracleTSC's explanations as structured, human-readable rationales that improve transparency and behavioral consistency, rather than as formal guarantees of internal causal computation.

Overall, the qualitative shift from hesitant exploration to structured and task-aligned rationales complements our quantitative improvements in traffic metrics, while acknowledging that reasoning faithfulness verification remains beyond the scope of the present work.

## 5 Discussion and Conclusion

This paper presents OracleTSC, a reinforcement learning framework for LLM-based TSC that explicitly targets the training instabilities inherent to long-horizon decision-making. The framework integrates two complementary mechanisms: (1) a reward hurdle that suppresses low-impact actions and sharpens long-term credit assignment, and (2) Temperature-scaled Softmax DSE, which regularizes uncertainty in semantic action generation and stabilizes the model's internal reasoning process across responses. Together, these components address fundamental challenges in applying PPO to LLM-based control, including reward sparsity, time-varying traffic patterns, and cross-token variance in semantic action selection.

We demonstrate that reinforcement fine-tuning of LLM controllers is brittle, and show that targeted reward shaping and uncertainty regularization substantially improve stability and performance. On CityFlow-$1 \times 1$, fine-tuned LLaMA3-8B reduces average travel time by 75% and queue length by 67%, with similar gains observed across multiple backbone models and intersection configurations. Importantly, OracleTSC performs competitively with specialized, black-box reinforcement learning controllers, while uniquely preserving

```
<think>
Okay, let's see. The current phase is ETEL. The intersection's state is given for each phase,
    and I need to figure out which signal is the most effective. </think>

First, I should understand what each signal does. The signal phases are ETL, NTST, ELWL,
    NLSL, ETEL, WTL, NTNL, STSL. Each phase allows certain lanes through and left-turn
    lanes. The goal is to choose the phase that best reduces delays, which would depend on
    the early queued vehicles and the number of approaching vehicles in different segments.

Looking at the data for each phase:

Phase ETL: Allowed lanes are Eastern through and left-turn. Early queued Eastern through: 3,
    Western through: 38. Segment 1: 1, segment 2: 3, segment 3: 13. Wait, the numbers might
    be different. Wait, the user provided the data for each phase. Let me check again.
Wait, the user's input for each phase's state is:

Phase:ETWT
Allowed lanes: Eastern and western through lanes.
Early queued Eastern through: 3.0 Western through: 38.0
segment_1 Eastern through: 1 Western through: 1
segment_2 Eastern through: 3 Western through: 3
segment_3 Eastern through: 13 Western through: 51

Wait, that seems conflicting. The initial state had Eastern through: 9, Western through: 38.
    Then in the second phase, ETL, it's 3 and 38. But the user's current input shows that
    after ETL, the next phase is ETL again. Wait, no, the user's message is:

The current phase is ETEL. The state of the intersection:

Phase:ETWT
Allowed lanes: Eastern and western through lanes.
Early queued Eastern through: 3.0 Western through: 38.0
segment_1 Eastern through: 1 Western through: 1
segment_2 Eastern through: 3 Western through: 3
segment_3 Eastern through: 13 Western through: 51

Wait, but in the previous interaction, the user had ETL, and then after that, the next phase
    is ETL again. So maybe the current phase is ETL, and the state for ETL is 3 and 38.

So for phase ETL, allowed lanes are through and left-turn. The early queued for through
    lanes is 3, and for left-turn is 0. The segments are 1, 3, 13 for through, and 0, 0, 0
    for left-turn. Wait, but the initial data had segment_1: 1, segment_2: 3, segment_3: 13.
    But in the next phase, maybe the segments change? Wait, no, the user's input here is for
     the current phase, so maybe the segments are as given.

So for phase ETL, the number of early queued vehicles in allowed lanes is 3 (through) and 0
    (left). The segments for through are 1, 3, 13. For left, they are 0.

Now, looking at other phases. For example, phase NTST: allowed lanes are northern and
    southern through. Early queued Northern through: 38, Southern through: 26. Segments for
    through: 0, 0, 38. So here, there's a high early queued in northern through.

Phase WTL: allowed lanes are western through and left-turn. Early queued: 38, 5. Segments:
    0, 0, 38...
```

Listing 1: Illustrative reasoning trace before fine-tuning. The model demonstrates self-contradictions and hesitation during phase selection (highlighted in **red**), with several phases misidentified through incorrect mnemonics (highlighted in **blue**), reflecting weak adherence to prompt instructions. In addition, certain phases (highlighted in **orange**) are repeatedly re-analyzed, inefficiently consuming the LLM's output token budget.

```
Step 1: Identify the optimal traffic signal by analyzing the current phase and its state. The
    current phase is ETEL, which allows Eastern through and left-turn lanes. The highest
    segment (segment 3) is 4, and there are no early queued vehicles. This suggests a
    potential for efficient traffic flow by prioritizing the farthest segments.

Step 2: The optimal traffic signal is ETEL. It maximizes segment utilization and minimizes
    delays by allowing vehicles to flow through the farthest segments without waiting for
    early queued vehicles.

<signal>ETEL</signal>
```

Listing 2: Output generation after finetuning with environmental, hurdle rate, and Temperature-scaled Softmax DSE

the interpretability and transparency afforded by natural-language decision rationales. Ablation studies further confirm that both the reward hurdle and Softmax DSE are critical to performance, with the strongest results obtained when the two mechanisms are jointly applied.

Despite these gains, several limitations remain. The optimal hurdle magnitude depends on environment dynamics and currently requires manual tuning, while semantic entropy estimation incurs additional computational cost due to multi-sample generation. These limitations motivate several future directions, including adaptive or learned hurdle scheduling, multi-agent coordination across networked intersections, and robustness evaluation under multimodal sensing and real-world deployment settings.

Importantly, our objective is not to claim superiority over all existing RL algorithms. Highly optimized, non-explainable RL controllers remain strong on several benchmarks. Rather, our goal is to improve the performance, stability, and reliability of explainable LLM-based controllers while preserving natural-language reasoning and semantic-level uncertainty modeling.

In summary, ORACLETSC demonstrates that combining reward filtering with uncertainty-aware semantic regularization provides a principled and effective pathway for stabilizing reinforcement fine-tuning of LLM-based controllers. These results suggest a broader role for uncertainty-controlled semantic objectives in long-horizon control tasks, bridging the gap between interpretability and performance in next-generation intelligent transportation systems.

**Acknowledgment**

This work was supported in part by the ORAU Ralph E. Powe Junior Faculty Enhancement Award, the National Science Foundation (NSF) NAIRR Start-Up Program under Award NAIRR250417, and the Lambda Research Grant. The authors gratefully acknowledge this support. Any opinions, findings, conclusions, or recommendations expressed in this paper are those of the authors and do not necessarily reflect the views of the supporting organizations.

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

# A   Appendix

**Contents**

### A.1 Reward Design and PPO Stabilization Sensitivity

This section provides additional empirical and methodological context for the reward design used in OracleTSC. Specifically, we examine the optimization behavior of PPO under the queue-length reward.

Let $\text{queue}_t$ denote the average number of vehicles whose speed falls below 0.1 m/s across all lanes. The standard congestion objective corresponds to the reward

$$R_q(s_O^t, a_O^t) = -\text{queue}_t \tag{15}$$

While this objective is well aligned with long-horizon traffic efficiency, in practice it produces weak action-conditioned gradients during short control intervals. Inflow variation and regime effects can dominate $\text{queue}_t$, producing large baseline shifts in reward that are not attributable to a single phase decision. This results in high-variance advantages and unstable policy updates when fine-tuning language-model controllers with PPO.

We evaluate whether standard PPO stabilization techniques suffice under $R_q$. Across multiple hyperparameter sweeps—including learning rate, GAE-$\lambda$, reward discount factor $\gamma$, and removal of the critic via REINFORCE—we observe either policy collapse or negligible improvements over the pretrained baseline.

Table 4: Learning Rate Sensitivity on CityFlow1x1 (Reward = Queue Length, Qwen3-0.6B)

| Learning Rate | Travel Time | Queue Length | Delay | Throughput |
|---|---|---|---|---|
| Baseline | 506.1 | 104.06 | 0.71 | 1485 |
| $1 \times 10^{-3}$ | 702 | 111.29 | 0.73 | 1256 |
| $1 \times 10^{-4}$ | 325 | 111.41 | 0.79 | 1749 |
| $1 \times 10^{-5}$ | 466.1 | 83.99 | 0.75 | 1568 |
| $1 \times 10^{-6}$ | 453.9 | 79.97 | 0.72 | 1568 |
| $1 \times 10^{-7}$ | 424.3 | 86.49 | 0.74 | 1576 |

Table 5: GAE-$\lambda$ Sensitivity on CityFlow1x1 (Reward = Queue Length, Qwen3-0.6B)

| $\lambda$ | Travel Time | Queue Length | Delay | Throughput |
|---|---|---|---|---|
| 0.7 | 490.9 | 87.86 | 0.72 | 1527 |
| 1.0 | 498.3 | 91.2 | 0.74 | 1513 |

Table 6: Reward Discount Factor Sensitivity on CityFlow1x1 (Reward = Queue Length, Qwen3-0.6B)

| $\gamma$ | Travel Time | Queue Length | Delay | Throughput |
|---|---|---|---|---|
| 0.000 | 538.8 | 94.99 | 0.71 | 1476 |
| 0.999 | 453.9 | 79.97 | 0.72 | 1568 |

These results indicate that optimizer-level tuning alone does not reliably stabilize learning under the absolute queue-length reward in our exchange-level fine-tuning regime.

### A.2 Hyper-Parameters for Table 1

The value estimator $V_\phi$ s parameterized as a two-layer feedforward network across all model families, sizes, and intersection configurations. The first hidden layer expands the input dimension from the model's hidden state size to twice that dimension, followed by a Leaky ReLU activation with negative slope 0.01. The second layer produces a scalar value estimate.

For optimization, we set the value estimator learning rate to $10^{-5}$ and weight decay to $5 \times 10^{-7}$ uniformly across experiments. Gradient clipping is applied with a maximum norm of 0.5 for the policy network and 5.0 for the value estimator.

Table 7: REINFORCE (No Critic, No Baseline) on CityFlow1x1 (Reward = Queue Length, Qwen3-0.6B)

|  | Travel Time | Queue Length | Delay | Throughput |
|---|---|---|---|---|
| Episode 1 | 440 | 104.03 | 0.75 | 1603 |
| Episode 2 | 628 | 75.19 | 0.61 | 1346 |
| Episode 3 | 664.7 | 94.38 | 0.68 | 1302 |
| Episode 4 | 716.9 | 89.47 | 0.69 | 1230 |

The KL divergence penalty uses the K3 approximation:

$$KL(p||q) = \frac{p}{q} - \log\frac{p}{q} - 1$$

weighted by $\beta = 0.05$ in the token-level reward. The policy is optimized using the clipped surrogate objective given in Equation 16.

Model-specific and intersection-specific hyperparameters are detailed in Table 8, where $\epsilon_l$ and $\epsilon_u$ denote the lower and upper PPO clipping bounds, and $H_R$ represents the reward hurdle threshold.

Table 8: Model-specific hyperparameters for experiments in Table 1

| Model | Intersection | Actor LR | Actor Weight Decay | $\epsilon_l$ | $\epsilon_u$ | $H_R$ |
|---|---|---|---|---|---|---|
| Qwen3-0.6B | CityFlow1x1 | $2.5 \times 10^{-5}$ | $10^{-6}$ | 0.2 | 0.5 | 3.7 |
| Qwen3-0.6B | Cologne1 | $2.5 \times 10^{-5}$ | $10^{-6}$ | 0.2 | 0.2 | 2.7 |
| Qwen3-8B | CityFlow1x1 | $3.0 \times 10^{-5}$ | $8 \times 10^{-7}$ | 0.2 | 0.2 | 3.1 |
| Qwen3-8B | Cologne1 | $2.5 \times 10^{-5}$ | $10^{-6}$ | 0.2 | 0.2 | 3.0 |
| LLaMA3-8B | CityFlow1x1 | $2.5 \times 10^{-5}$ | $10^{-6}$ | 0.2 | 0.5 | 3.0 |
| LLaMA3-8B | Cologne1 | $2.5 \times 10^{-5}$ | $10^{-6}$ | 0.2 | 0.2 | 3.7 |

### A.3 Supported Phase Mnemonics for Evaluated Intersections

Table 9: Supported Phase Mnemonics for CityFlow1x1.

| Phase Mnemonic | Description |
|---|---|
| NTST | Northern and southern through lanes |
| NLSL | Northern and southern left-turn lanes |
| NTNL | Northern through and left-turn lanes |
| STSL | Southern through and left-turn lanes |
| ETWT | Eastern and western through lanes |
| ELWL | Eastern and western left-turn lanes |
| ETEL | Eastern through and left-turn lanes |
| WTWL | Western through and left-turn lanes |

Table 10: Supported Phase Mnemonics for Cologne1.

| Phase Mnemonic | Description |
|---|---|
| NUTRLSUTRL | Northern and southern U-turn, through, right-turn and left-turn lanes |
| NUTLSUTL | Northern and southern U-turn, through, and left-turn lanes |
| EUTRLWUTRL | Eastern and western U-turn, through, right-turn and left-turn lanes |
| EUTLWUTL | Eastern and western U-turn, through, and left-turn lanes |

## A.4 System Prompt

```
You are an expert in traffic management. You can use your knowledge of traffic commonsense
    ↪ to solve traffic signal control tasks.
A traffic light regulates a four-section intersection with northern, southern, eastern, and
    ↪ western sections, each containing two lanes: one for through traffic and one for
    ↪ left-turns. Each lane is further divided into three segments.
Segment 1 is the closest to the intersection. Segment 2 is in the middle. Segment 3 is the
    ↪ farthest. In a lane, there may be early queued vehicles and approaching vehicles
    ↪ traveling in different segments. Early queued vehicles have arrived at the
    ↪ intersection and await passage permission. Approaching vehicles will arrive at the
    ↪ intersection in the future.
The traffic light has 8 signal phases. Each signal relieves vehicles' flow in the two
    ↪ specific lanes. The state of the intersection is listed below. It describes:
- The group of lanes relieving vehicles' flow under each signal phase.
- The number of early queued vehicles of the allowed lanes of each signal.
- The number of approaching vehicles in different segments of the allowed lanes of each
    ↪ signal. The signal phase you choose will persist for 10 seconds.
Please answer: Which is the most effective traffic signal that will most significantly
    ↪ improve the current traffic condition?
Requirements:
- Let's think step by step.
- You can only choose one of the signals listed by the user.
- You must follow the following steps to provide your analysis:
   Step 1: Provide your analysis for identifying the optimal traffic signal.
   Step 2: Answer your chosen signal.
- Your choice can only be given after finishing the analysis. - Your choice must be
    ↪ identified by the tag: <signal>YOUR_CHOICE</signal>.
```

Listing 3: Our System Prompt

## A.5 Textual Phase Representation

```
Phase: ETWT (Eastern and Western Through Lanes)
**Early Queued Vehicles**:
 - Eastern through: 17
 - Western through: 3
**Approaching Vehicles**:
 - Segment 1: Eastern through: 1, Western through: 2
 - Segment 2: Eastern through: 5, Western through: 7
 - Segment 3: Eastern through: 17, Western through: 20
```

Listing 4: Representation of a phase and its traffic observation. Example shown is for Eastern and Western Through (ETWT) lanes in CityFlow1x1.

## A.6 Algorithms

---

**Algorithm 1** PPO-Based Training for LLM-Guided Traffic Signal Control using Hurdle Rate and Softmax with Temperature Discrete Semantic Entropy

---

**Require:** Number of responses for entropy computation $G$, entropy weight $\alpha$, clipping threshold $\epsilon$, discount $\gamma$, GAE parameter $\lambda$, episode length $T$, maximum number of output tokens $L$, boolean symbolizing whether to use Discrete Semantic Entropy with or without Softmax use_softmax, *optional* Softmax Temperature $\tau$, number of batches to train over $n_b$, batch size $B$, Hurdle Rate $H_R$

1: Initialize Buffer $\mathcal{B} \leftarrow \emptyset$
2: Load pretrained language model $\pi_\theta$ and reference model $\pi_{\text{ref}}$
3: Initialize value head $V_\phi$
4: **for** each training iteration **do**
5:    **for** each environment step $t = 1$ to $T$ **do**
6:       Observe traffic state $s$ and current signal phase $p$
7:       Create verbalized traffic state $s_0^t$ using $s$, $p$
8:       Sample response $\{a_0^t, \ldots, a_O^t\} \sim \pi_\theta(\cdot \mid s_0^t)$ with maximum length $O$
9:       Parse selected action $p_{next}$ from $a_t$
10:      Sample $G$ responses $\{\{a_0^t, \ldots, a_O^t\} \sim \pi_\theta(\cdot \mid s_t)\}_{g=1}^G$ with maximum length $O$
11:      Compute phase histogram $\{\text{count}_p^{\text{answer}}\}_{p=1}^{N_P}$ from $G$ responses using Algorithm 3
12:      **if** use_softmax **then**
13:         Compute $r_E \leftarrow R^{\text{answer}}$ from $\{\text{count}_{p'}^{\text{answer}}\}_{p'=1}^{N_P}$, $p_{next}$, $\tau$ using Equation 11
14:      **else**
15:         Compute $r_E \leftarrow R_E^{\text{answer},DSE}$ from $\{\text{count}_{p'}^{\text{answer}}\}_{p'=1}^{N_P}$, $p_{next}$ using Equation 13
16:      **end if**
17:      Execute action $p_{next}$, observe reward $R(s_O^t, a_O^t)$, next state $s_0^{t+1}$
18:      Compute token-level rewards $r_t$ from total reward $R_{\text{hurdle}}(s_O^t, a_O^t)$ using Equations 14 and 1
19:      Compute action sequence log-probabilities $\{\log \pi_{\theta_{old}}(a_o^t \mid s_{o-1}^t)_{o=1}^O\}$ for all output tokens in $a_0^t, \ldots, a_O^t$ from $\pi_\theta$.
20:      Compute state value estimate $V_{old} \leftarrow V_\phi(s_0^t)$.
21:      Store $(s_0^t, a_t = \{a_0^t, \ldots, a_O^t\}), r_t,$ $\log \pi_{\theta_{old}}(a_t \mid s_t), V_{old})$ in buffer $\mathcal{B}$
22:    **end for**
23:    Compute Advantages $\hat{A} = [\hat{A}_t \quad \forall t \in [1, T]]$ using Generalized Advantage Estimation($r_1, \ldots, r_T, V_\phi(s_1), \ldots, V_\phi(s_T), \gamma, \lambda$).
24:    Compute $\lambda$-Return $\hat{R} = [\hat{A}_t + V_\phi(s_t) \quad \forall t \in [1, T]]$.
25:    Shuffle Buffer $\mathcal{B}, \hat{R}, \hat{A}$ in-place.
26:    Get $n_b$ batches from $\mathcal{B}, \hat{A}, \hat{R}$ of batch size $B$ each.
27:    **for** $b \in [1, n_b]$ **do**
28:      $(s^{\text{batch}}, a^{\text{batch}}, r^{\text{batch}}, \log \pi_{old}^{\text{batch}}, V_{old}^{\text{batch}}),$ $\hat{A}^{\text{batch}}, \hat{R}^{\text{batch}} \leftarrow \text{batch}_b$
29:      Standardize all elements in $\hat{A}^{\text{batch}}$ in-place.
30:      Compute action sequence log-probability $\log \pi_\theta^{\text{batch}}$ for all output tokens in $a^{\text{batch}}$.
31:      Compute ratio: $\text{ratio}^{\text{batch}} = \exp(\log \pi_\theta^{\text{batch}} - \log \pi_{\text{old}}^{\text{batch}})$
32:      Compute Policy objective at batch-level using Equation 16.
33:      Compute Value objective at batch level using Equation 5.
34:      Update $\theta, \phi$ with $\nabla \mathcal{J}(\theta, \phi)$ using Equation 6
35:    **end for**
36:    # Clear buffer
37:    Set $B \leftarrow \emptyset$
38: **end for**

---

---

**Algorithm 2** Extract Phase from Output Text

---

**Require:** Output text $s$, valid phase mnemonics $\mathcal{K}$, phase descriptions $\mathcal{D}$, mapping from phase mnemonics to OpenAI Gym action classes `phase_to_code`, default action class `default_code` $\in [0, N_P)$
  1: Extract all matches $\mathcal{M} \leftarrow$ regex search for `<signal>...</signal>` in $s$
  2: **for** phase $\in$ reverse($\mathcal{M}$) **do**
  3:   **if** phase $\in \mathcal{K}$ **then**
  4:     **return** `phase_to_code`[phase]
  5:   **end if**
  6: **end for**
  7: $s_{low} \leftarrow$ lowercase($s$), $i \leftarrow -1$, $k^* \leftarrow$ None
  8: **for all** $k \in \mathcal{K}$ **do**
  9:   $j_k \leftarrow$ `rfind`($s_{low}$, $k$.`lower()`)
 10:   $j_d \leftarrow$ `rfind`($s_{low}$, $\mathcal{D}[k]$.`lower()`)
 11:   **if** $\max(j_k, j_d) > i$ **then**
 12:     $k^* \leftarrow k$, $i \leftarrow \max(j_k, j_d)$
 13:   **end if**
 14: **end for**
 15: **if** $k^* \neq$ None **then**
 16:   **return** `phase_to_code`[$k^*$]
 17: **else**
 18:   **return** `default_code`
 19: **end if**

---

---

**Algorithm 3** Compute Unnormalized Probability of Phase from LLM Responses

---

**Require:** Number of responses $G$, Policy LLM $\pi_\theta$, Input state $s$
**Ensure:** Histogram of extracted phases from $G$ responses, $h$
  1: Generate $G$ output responses: $\{a_1, a_2, \ldots, a_G\} \leftarrow \pi_\theta(s)$
  2: Initialize empty list $\mathcal{P} \leftarrow []$
  3: **for** $g = 1$ to $G$ **do**
  4:   $p_g \leftarrow$ `ExtractPhase`($a_g$)                                    {See Algorithm 2}
  5:   Append $p_g$ to $\mathcal{P}$
  6: **end for**
  7: Build histogram $[\text{count}_0^{\text{answer}}, \ldots, \text{count}_{N_P-1}^{\text{answer}}]$ over $\mathcal{P}$ with $N_P$ bins for $N_P$ supported phases
  8: **return** Phase Histogram $[\text{count}_0^{\text{answer}}, \ldots, \text{count}_{N_P-1}^{\text{answer}}]$

---

### A.7 Ablation on Upper Clip Limit

We ablate the effect of the **upper clip limit** $\epsilon_u$ in the clipped surrogate objective (Equation 16) to examine how it influences the policy stability and learning dynamics of language-model-based controllers. In traditional PPO, a smaller clip range $(1 \pm \epsilon)$ constrains the policy update and prevents destructive policy shifts, while a larger range permits more aggressive updates but risks over-fitting to noisy or stochastic rewards. In our entropy-controlled formulation, $\epsilon_u$ further interacts with the uncertainty-weighted auxiliary reward $w_{\text{uncertainty}} R_E^{\text{answer}}$, effectively scaling the policy's sensitivity to exploration signals. Prior work Shen (2025) suggests that larger upper clip limits can sustain higher entropy across training, discouraging premature convergence to deterministic policies.

As shown in Figures 11 and 12, this trade-off manifests clearly in both models. For **Qwen3-0.6B** on the Cologne1 intersection ($H_R = 2.7$), increasing $\epsilon_u$ from 0.2 to 0.5 led to degraded traffic efficiency, reflected in higher travel time and queue length. The smaller clip coefficients yielded smoother delay curves and higher throughput, indicating more stable and well-regularized learning. We observe a similar pattern with **Qwen3-8B** on CityFlow1x1 ($H_R = 3.1$); strictest clipping ($\epsilon_u \approx 0.2$) achieved the best balance between

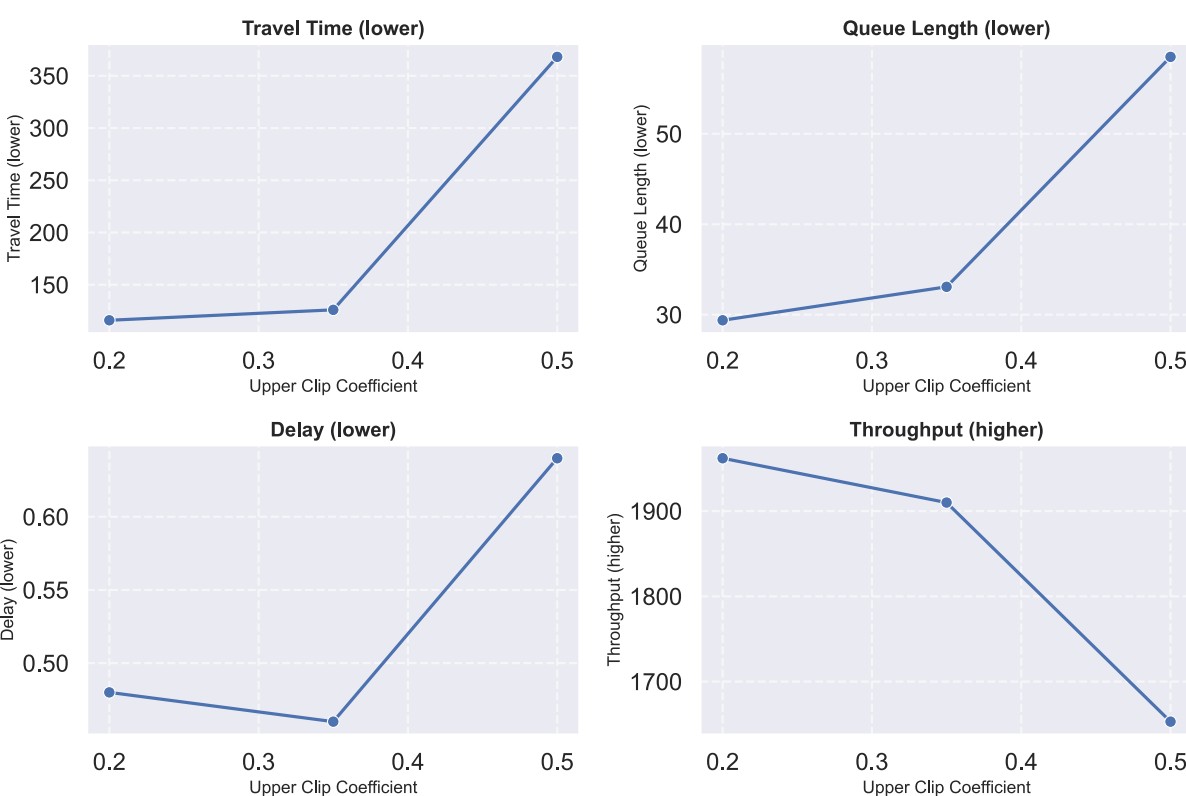

Figure 11: Effect of the Upper Clip Limit $\epsilon_u$ in the $R(s_O^t, a_O^t) - H_R + w_{\text{uncertainty}} R_E^{\text{answer}}$ configuration where $w_{\text{uncertainty}} = 1$ and $H_R = 2.7$. Each subplot reports one evaluation metric (Travel Time, Queue Length, Delay, and Throughput) as a function of $\epsilon_u$ for Qwen3-0.6B on the Cologne1 intersection.

exploration and control, while larger clip limits again inflated travel time and reduced throughput. However, Delay metrics performed slightly better with a marginally higher $\epsilon_u = 0.35$.

Overall, these results reinforce the importance of **bounded update magnitudes** in language-model-based reinforcement learning. Tight upper clip limits constrain policy drift and encourage gradual adaptation, enabling the agent to maintain interpretable, state-responsive decision patterns rather than collapsing to single-phase policies. However, excessively restrictive clipping can also suppress meaningful exploration. Hence, an intermediate range of $\epsilon_u$ appears most effective—sufficiently permissive to preserve entropy yet conservative enough to maintain reward stability.

$$\mathcal{J}_{\text{CLIP}}(\theta) = \hat{\mathbb{E}}_{\tau \sim \pi_{\text{old}}} \Big[ \min\big(r_l(\theta)\, \hat{A}_l,\ \text{clip}(r_l(\theta),\, 1 - \epsilon_l,\, 1 + \epsilon_u)\, \hat{A}_l\big)\Big], \tag{16}$$

## A.8 Experimental Rigor: Repeating with Multiple Seeds

To improve experimental robustness, we evaluate the primary configuration across multiple random seeds.

Across seeds, performance remains stable. Travel time varies within a narrow band (140.2–243.5), and throughput remains consistently high (1844–1975). Queue length shows moderate variation (42.93–82.65), but no run exhibits policy collapse or severe degradation.

**Qwen3-8B (CityFlow1x1) – Upper Clip Ablation (Hurdle = 3.1)**

Figure 12: Effect of the Upper Clip Limit $\epsilon_u$ in the $R(s_O^t, a_O^t) - H_R + w_{\text{uncertainty}} R_E^{\text{answer}}$ configuration where $w_{\text{uncertainty}} = 1$ and $H_R = 3.1$. Each subplot reports one evaluation metric (Travel Time, Queue Length, Delay, and Throughput) as a function of the $\epsilon_u$ for Qwen3-8B on the CityFlow1x1 intersection.

Table 11: Performance across three independent random seeds on CityFlow1x1 (Reward = Queue Difference − Hurdle Rate + Temperature-scaled Softmax DSE, LLaMA3-8B)

| Seed | Travel Time | Queue Length | Delay | Throughput |
|------|-------------|--------------|-------|------------|
| 0    | 146.9       | 45.5         | 0.67  | 1946       |
| 11   | 173.2       | 53.65        | 0.66  | 1975       |
| 6446 | 243.5       | 82.65        | 0.66  | 1844       |

These results suggest that the observed gains are not attributable to a single favorable initialization. We acknowledge that larger-scale seed evaluation would provide tighter confidence intervals. However, each full fine-tuning run requires approximately 12 hours of GPU time under our four-episode training protocol. Given the computational cost of long-horizon LLM reinforcement learning, large multi-seed sweeps are uncommon in recent LLM-based traffic signal control studies Lai et al. (2025b); Wang et al. (2024a). Nonetheless, the additional runs presented here strengthen the empirical validity of our findings.

### A.9    Ablation on Number of Responses (G)

Table 12: Sensitivity to the Number of Responses ($G$) on CityFlow1x1 (Reward = Queue Difference − Hurdle Rate + Temperature-scaled Softmax DSE, LLaMA3-8B). (Episode 2)

| Number of Responses (G) | Travel Time | Queue Length | Delay | Throughput |
|:---:|:---:|:---:|:---:|:---:|
| 2 | 377 | 82.04 | 0.73 | 1665 |
| 4 | 315.5 | 103.86 | 0.76 | 1750 |
| 8 | 146.9 | 45.5 | 0.67 | 1946 |

As $G$ increases, performance improves consistently in terms of travel time and throughput. Increasing $G$ strengthens semantic-level uncertainty estimation by aggregating agreement across multiple independently sampled reasoning trajectories. The $G = 8$ configuration achieves the strongest results, reducing travel time by approximately 41% compared to $G = 2$, and increasing throughput by over 12%.

However, this improvement comes at substantial computational cost. Because each additional response requires a full forward rollout and reward evaluation, GPU memory usage scales approximately linearly with $G$.

These results highlight a clear performance–efficiency trade-off. Smaller values of $G$ reduce computational burden but weaken semantic consensus regularization, while larger values improve stability and final policy quality at increased cost. In practice, $G$ should be selected based on available compute budget and desired stability guarantees.

