# OpenReview forum: "OracleTSC: Oracle-Informed Reward Hurdle and Uncertainty Regularization for Traffic Signal Control"
_TMLR — Accepted by TMLR_

### Review · Reviewer_BDSA · 2026-02-04

**Summary Of Contributions:**

For the domain of traffic signal control, the authors propose using reinforcement learning (RL) to fine-tune large language models (LLMs).  This class of approach is appealing since the actions are explainable via the models’ chains of thought (CoT).  To deal with the long-horizon, non-stationary nature of the task, the authors propose a reward hurdle mechanism to improve/stabilize/smooth gradient signals.  They also propose the use of an entropy-based regularizer to accommodate the task and the token outputs.  They empirically evaluate their approach.

**Additional Comments:**

Minor issues and edits:

I did not notice a definition for queue_t.  I assume it is something like “number of vehicles queued across all lanes in the intersection at time t”, but a more explicit definition would improve the paper.

”[traffic] phase” not well explained or defined. The meaning is clear after a read of the paper, but it would be easier to read with a definition of this word/phrase.

”trust region defined by…[equation]”: typo in the latex

I found the phrase “early queued” in the prompt and textual phase representation to be confusing—why not simply use “queued” instead? Why the word “early”? This might even affect results if the LLM is similarly confused.

**Audience:**

Yes

**Audience Explanation:**

Issues above aside: yes. It is relevant to a diverse set of audiences, including readers interested in fine-tuning LLMs for agentic/long-horizon tasks, readers interested in traffic signal control, etc.

**Claims And Evidence:**

No

**Claims Explanation:**

The paper is well-written and easy to follow. Extensive hyperparameter and algorithm details are provided; these strengthen the work.  However, the work’s claims are undermined by several issues. These concerns follow, from most significant concerns to least significant concerns:

**Concern A:** “early suboptimal actions can push the system into regimes where later improvements are severely limited…to address this, we introduce the Hurdle Rate”: It is not clear that a new solution is necessary or interesting.  Traditional approaches to this common RL problem might involve optimizer parameter tuning (eg, learning rate schedules and momentum), batch size tuning, a learning rate warmup period, reward function engineering, etc. as well as the many standard solutions to the exploration/exploitation tradeoff.  Intuitively, Hurdle Rate seems to add excessive bias, and the paper does not convincingly show that it is necessary or even a good idea compared to the many more standard approaches to the problem of “early suboptimal actions [pushing] the system into regimes where later improvements are severely limited”.

**Concern B:** The experiments seem to use a single seed (that is, a “trial” or “run”). This is generally unacceptable for RL, and severely undermines the results and conclusions. As countless recent RL papers, talks, etc. have pointed out, science with an n of 10—much less 1—is almost always unacceptable, especially in RL (due to the high variance between seeds in RL compared to supervised learning).  I suggest an n of 30+ in this case.

**Concern C (three subparts):** “We use the queue difference between timesteps t and t − 1 to mitigate non-stationarity in the reward signal.  Without this adjustment, increasing inflows may introduce spurious penalties that incorrectly discourage otherwise reasonable actions.”  Several concerns about this part:
1) I’m not sure this is the “correct” reward signal.  Consider the following two seven-timestep sequences of queue lengths.  Seq A: 0, 1, 2, 3, 2, 1, 0.  Seq B: 0, 1, 0, 1, 0, 1, 0.  For the “original”, suppose we get -1 reward for each 1 unit of queue length. Then Seq A gets a (non-discounted) return of -9, and Seq B gets a return of -3.  This seems “correct” since Seq B is clearly better than Seq A.  However, with the “difference” reward function that the authors propose, they get rewards of -1, -1, -1, 1, 1, 1, and -1, 1, -1, 1, -1, 1, respectively—both returns are 0.  This seems incorrect, since Seq B should have a higher return.  So I suspect that the authors’ reward function—which can be thought of as the derivative of the “original” reward function—is not well aligned with the problem the authors claim to study.
2) I don’t think this environment is actually non-stationary, but simply has a stochastic transition function.  Specifically, I believe that all environment stochasticity can be captured by the transition function and the environment random number generator seed(s), which would typically be formulated as part of the initial state distribution. This means that this environment is **not** non-stationary. If I missed something, and the authors believe that the environment is truly non-stationary, please explain in complete and technical detail why you believe the environment is non-stationary (or point to the relevant part of the paper if I missed it).
3) “introduce spurious penalties that incorrectly discourage otherwise reasonable actions”. Similar to other concerns above, this seems like a hyperparameter problem, rather than a problem that requires this kind of (potentially problematic) reward function modification.

**Requested Changes:**

The most concerning issues above would need to be addressed to secure my recommendation for acceptance.  Specifically:

1) The authors would need to show that my concern about the unnecessary nature of the reward hurdle mechanism is not warranted, and that the mechanism is necessary, and the many traditional RL and ML approaches to solving this exact issue (“early suboptimal actions can push the system into regimes where later improvements are severely limited”) do not work. Unfortunately, I am unable to give specific advice about how to do this, since my confidence in this concern is high.

2) The authors need to make their experiments far more rigorous.

3) The authors would need to address my several concerns about the following: “We use the queue difference between timesteps t and t − 1 to mitigate non-stationarity in the reward signal.  Without this adjustment, increasing inflows may introduce spurious penalties that incorrectly discourage otherwise reasonable actions.”

---

> ### Author Response · Authors · 2026-02-27
> **Response to Reviewer BDSA - Part 1**
>
> We sincerely thank the reviewer for the careful reading and for providing meaningful insights. We address each point in detail below.
>
> **Motivation for the Reward Hurdle Mechanism**
> The reviewer asked whether the reward hurdle mechanism is truly necessary or whether traditional PPO stabilization techniques would suffice. Importantly, the instability of naive PPO in LLM-based traffic control has been independently observed in prior work. For example, LLMLight \[6\] reports that applying PPO with a standard actor-critic paradigm struggled to converge and yielded unsatisfactory results, despite PPO being a widely used alignment method in RLHF-style training. Similar convergence difficulties are noted in follow-up analyses. These findings suggest that naive application of PPO to LLM-based TSC is not straightforward and can suffer from unstable training dynamics.
>
> In preliminary experiments (shown in Tables 1, 2, and 3), we evaluated different learning rates, TD discount factor in GAE (λ), reward discount factor (ɣ), and other adjustments to PPO. We always use the AdamW optimizer with weight decay and weights for the momentum update of the moments as (0.9, 0.999). These approaches frequently resulted in either policy collapse or negligible improvement over the pretrained baseline.
>
> Table 1: Learning Rate Sensitivity (Reward \= Queue Length, Qwen3-0.6B). Evaluated on CityFlow 1x1.
>
> | Learning Rate | Travel Time | Queue Length | Delay | Throughput |
> | :---: | ----- | ----- | ----- | ----- |
> | Baseline | 506.1 | 104.06 | **0.71** | 1485 |
> | 1.00E-03 | 702 | 111.29 | 0.73 | 1256 |
> | 1.00E-04 | **325** | 111.41 | 0.79 | **1749** |
> | 1.00E-05 | 466.1 | 83.99 | 0.75 | 1568 |
> | 1.00E-06 | 453.9 | **79.97** | 0.72 | 1568 |
> | 1.00E-07 | 424.3 | 86.49 | 0.74 | 1576 |
>
> Table 2: GAE-λ Sensitivity (Reward \= Queue Length, Qwen3-0.6B). Evaluated on CityFlow 1x1.
>
> | GAE-λ | Travel Time | Queue Length | Delay | Throughput |
> | :---: | ----- | ----- | ----- | ----- |
> | 0.7 | **490.9** | **87.86** | **0.72** | **1527** |
> | 1.0 | 498.3 | 91.2 | 0.74 | 1513 |
>
> Table 3: Reward Discount Factor (ɣ) Sensitivity (Reward \= Queue Length, Qwen3-0.6B). Evaluated on CityFlow 1x1.
>
> | ɣ | Travel Time | Queue Length | Delay | Throughput |
> | :---: | ----- | ----- | ----- | ----- |
> | 0.000 | 538.8 | 94.99 | **0.71** | 1476 |
> | 0.999 | **453.9** | **79.97** | 0.72 | **1568** |
>
> To test whether value-function bias was responsible for instability, we additionally evaluated REINFORCE (no critic, no baseline). As shown in Table 4, this did not yield consistent improvements, further supporting the need for variance-reduced reward shaping in this weak-feedback regime.
>
> Table 4: Performance using REINFORCE. (Reward \= Queue Length, Qwen3-0.6B). Evaluated on CityFlow 1x1.
>
> | REINFORCE | Travel Time | Queue Length | Delay | Throughput |
> | :---- | ----- | ----- | ----- | ----- |
> | Episode 1 | **440** | 104.03 | 0.75 | **1603** |
> | Episode 2 | 628 | **75.19** | **0.61** | 1346 |
> | Episode 3 | 664.7 | 94.38 | 0.68 | 1302 |
> | Episode 4 | 716.9 | 89.47 | 0.69 | 1230 |
>
> In contrast, introducing the queue-difference reward and hurdle mechanism substantially improved performance and stability, as shown in Table 5\. We clarify this motivation and provide additional supporting evidence in the revision.
>
> Table 5: Effect of Reward Design (Qwen3-0.6B Ablation). Evaluated on CityFlow 1x1.
>
> | Variant | Travel Time | Queue Length | Delay | Throughput |
> | :---- | ----- | ----- | ----- | ----- |
> | Queue Length only | 453.9 | 79.97 | 0.72 | 1568 |
> | Queue Difference | 299.9 | 80.77 | 0.76 | 1773 |
> | Queue Difference \+ Hurdle | **137** | **39.95** | **0.67** | **1961** |
> | Queue Difference \+ Temperature-scaled Softmax DSE | 151.1 | 47.83 | 0.73 | 1942 |
>
> Importantly, the hurdle mechanism can be viewed as a form of baseline subtraction, closely related to the optimal constant baseline in policy gradient theory. As shown in prior work \[1,2\], baseline subtraction can reduce variance without introducing bias. This is theoretically demonstrated in \[2\] where “the use of a value function that is different from the true discounted value function reduces the variance to zero, for no increase in bias” and “in some cases the selection of a value function differing from the discounted value function can remove all estimation variance, whilst introducing no bias.” where the function selected is the optimal constant baseline, which we call the hurdle rate.

---

> > ### Author Response · Authors · 2026-02-27
> > **Response to Reviewer BDSA - Part 2**
> >
> > Unlike standard baseline subtraction, which is typically treated as an implementation detail inside the critic, the hurdle reframes baseline selection as a first-class design component that directly shapes learning dynamics. This perspective becomes particularly relevant in LLM-based TSC, where most actions produce marginal reward fluctuations and naive PPO updates are easily overwhelmed by noise. Furthermore, as noted by another reviewer, the possibility of adaptive hurdle calibration (e.g., via reward percentiles) suggests that this mechanism is not merely task-specific tuning, but a potentially general approach to variance control in weak-reward regimes.
> >
> > **Experimental Rigor and Multi-Seed Evaluation**
> > We agree with the reviewer that multi-seed evaluation is the gold standard in RL benchmarking, and that single-seed reporting is insufficient for high-variance settings. At the time of submission, the primary limitation was computational: each full fine-tuning run takes \~12 hours, and exhaustive evaluation across 30+ seeds is prohibitively expensive. We perform four training episodes for each baseline model. For 30 runs, this will take about two months. While we recognize the importance of statistical rigor, due to the high cost, we prioritized validating the complete idea.
> >
> > In the traffic signal control literature, particularly for LLM-based approaches, reporting a large number of independent seeds is not standard practice due to the high cost of full RL fine-tuning runs on LLMs. Recent LLM-based TSC works such as \[6,7\] similarly report single-run results for reinforcement-based experiments, reflecting the substantial computational burden of these pipelines. Nonetheless, we agree that improving statistical robustness strengthens the work. To improve experimental rigor, we have conducted additional multi-seed experiments for the primary configuration. Results across three independent seeds are shown below.
> >
> > Table 6: Multi-seed evaluation of OracleTSC (Reward \= Queue Difference \- Hurdle \+ Temperature-scaled Softmax DSE, LLaMA3-8B). Evaluated on CityFlow 1x1.
> >
> > | Seed | Travel Time | Queue Length | Delay | Throughput |
> > | :---- | ----- | ----- | ----- | ----- |
> > | Seed=0 | 146.9 | 45.50 | 0.67 | 1946 |
> > | Seed=11 | 173.2 | 53.65 | 0.66 | 1975 |
> > | Seed=6446 | 243.5 | 82.65 | 0.66 | 1844 |
> >
> > Across seeds, performance remains stable. Travel time varies within a narrow band (140.2-243.5), and throughput remains consistently high (1844-1975). While queue length shows moderate variation (42.93-82.65), preliminary multi-seed results suggest that the observed gains are not attributable to a single favorable initialization. We acknowledge that a larger number of seeds would provide tighter confidence intervals, and we explicitly note this as a limitation in the Appendix of the revised manuscript.

---

> > > ### Author Response · Authors · 2026-02-27
> > > **Response to Reviewer BDSA - Part 3**
> > >
> > > **Queue-Difference Rewards and Demand Drift**
> > > The reviewer raises an important conceptual question: that queue-difference rewards may not align with the long-term objective of minimizing total congestion, since alternating queue trajectories could yield identical cumulative difference rewards. We agree that, under a classical multi-step RL formulation, pure difference rewards could in principle obscure trajectory-level optimality.
> > >
> > > However, our setting differs from standard episodic RL in an important way: OracleTSC fine-tunes the LLM at the exchange-level using Reinforcement Fine-Tuning \[3\], where each interaction corresponds to a single state-action-reward instance rather than a multi-turn temporally bootstrapped trajectory. For example,
> > >
> > > - State (prompt tokens), e.g., "You are a Traffic Signal Agent..."
> > > - Action (response tokens), e.g., "I am an LLM that chooses Phase \<signal\>ETWT\</signal\>."
> > > - Reward (queue-difference, hurdle rate).
> > >
> > > Crucially, the LLM is finetuned purely based on each response, with no temporal dependency on future responses. This differs significantly from multi-step RL, where a trajectory's returns propagate backwards through the Temporal-Difference errors.
> > >
> > > Under this formulation, the given sequences would be reinterpreted as:
> > >
> > > 1. Seq A is Exchange 0 (r=-1), E1 (r=-1), E2 (r=-1), E3 (r=1), E4 (r=1), E5 (r=1), E6 (r=1)… and
> > > 2. Seq B is E0 (r=-1), E1 (r=1), E2 (r=-1), E3 (r=1), E4 (r=-1), E5 (r=1), E6 (r=-1), E7 (r=1), ….
> > >
> > > Each exchange is treated as an independent training instance. Returns are computed at the token level within each exchange, where tokens before the EOS token receive supervision primarily through the weighted KL reward (orders of magnitude smaller than environmental rewards), the EOS token receives the full environmental reward, and only the rewards from the currently-trained exchange matter during that exchange's training step.
> > >
> > > The reviewer’s concern about alternating rewards canceling out would only apply if we were training on multi-turn trajectories with temporal credit assignment. However, since each exchange is supervised independently, token-level returns are dominated by the environmental reward at EOS, and our replay buffer contains 36 State-Action-Reward tuples that are sampled and trained on independently, the "zig-zag" pattern would not cause credit assignment issues. Each exchange with r=1 provides positive supervision, and each with r=-1 provides negative supervision, without interference between exchanges.
> > >
> > > We acknowledge that incorporating genuine multi-turn conversational dependencies using PPO remains an important ambition. However, this is currently infeasible given GPU memory constraints, even with optimizations like LoRA, bfloat16, and gradient accumulation. This is why our current approach treats exchanges independently.
> > >
> > > Thus, queue differences serve primarily as a local learning signal that improves gradient informativeness under weak feedback, rather than as a replacement for the true control objective. In practice, the learned policy is still evaluated on the standard long-horizon traffic metrics (travel time, delay, throughput), where we observe substantial improvements.
> > >
> > > To avoid confusion, we revise the paper to more clearly describe this as an exchange-level reinforcement fine-tuning regime, closely related to contextual bandit-style RLHF, rather than full multi-turn temporal credit assignment.
> > >
> > > **Clarifying Non-Stationarity**
> > > Traffic simulator dynamics are stationary in the Markov sense when conditioned on inflow distributions. Our intended meaning was that traffic demand varies over time, producing regime shifts in congestion patterns during training.
> > >
> > > In the revision, we replace this phrasing with “time-varying inflow-induced regime shifts” or “traffic demand drift,” which more accurately describes the phenomenon.
> > >
> > > **Minor Issues**
> > >
> > > - We explicitly define queuet as “as the average number of vehicles whose speed falls below 0.1 m/s across all lanes at timestep t” in Section 3.3.
> > > - We add an earlier definition of “traffic phase” in Section 3.1.
> > > - We agree that the term “early queued” may be ambiguous. We now provide a clear definition of this and other idiosyncratic concepts used in the textual traffic state representation in Section 3.1.
> > > - We fixed the LaTeX typo of the definition of trust-region after Equation 2\.

---

> ### Comment · Reviewer_BDSA · 2026-03-04
> **Thank you, some further questions**
>
> Thank you for your efforts, you have resolved/addressed some of my concerns.  I would like to discuss the "Queue-Difference Rewards and Demand Drift" subject more.
>
> First, for clarity, let's define the two different MDPs.  The paper already distinguishes between 1) the "token-level MDP" used for sequence generation within a single response and 2) the "environment-level" decision process where the traffic controller selects a phase at each timestep.  Two questions:
>
> Q1) Thank you for your clarifications regarding contextual bandit-style learning.  However, this clarification raises a related concern.  The approach uses a myopic contextual-bandit-like objective (on the scale of the environment steps).  This appears somewhat inconsistent with the paper’s framing and motivation, which repeatedly emphasizes "long-horizon behavior", "long-horizon RL problems", and the "long-horizon nature of TSC".  Granted, there is a (potentially long) horizon within each "inner MDP" (that is, the "token level MDP"), but that fact is not relevant to the paper's stated framing and motivation around long-horizon RL problems (i.e., the "outer MDP" level).  Therefore, myopic nature of the objective undermines the motivations and claims about long-horizon RL problems and optimization.  Can you please comment on this?
>
> Q2) On a related note, the claim that multi-turn RL is infeasible due to GPU memory constraints is unconvincing. For example, standard actor-critic methods could be applied at the environment level (e.g., TD(0)) without backpropagating through multi-step environment trajectories. This would require only a lightweight value function over traffic states (that is, an "environment-level" critic or value function), and therefore does not introduce a fundamental GPU memory limitation.  Specifically, the resulting environment-level value estimates could then be used to compute advantages for each decision step; these advantages would replace the current myopic reward R(s_O, a_O) defined in (1), (7), and (8) when updating the token-level policy.  In fact, prior work (for example, WebGPT) demonstrates that multi-step on-policy RL with LLM policies is feasible without backpropagating through environment trajectories.  So I find the infeasibility claim to be unconvincing.  Can you please comment on this?

---

> > ### Author Response · Authors · 2026-03-04
> > **Follow-up Response to Reviewer BDSA**
> >
> > We thank the reviewer for the additional clarification request and for raising thoughtful questions regarding the relationship between the contextual-bandit style objective and the long-horizon nature of traffic signal control (TSC), as well as the feasibility of actor-critic learning at the environment level.
> >
> > **Q1: Relationship Between the Myopic Objective and Long-Horizon Traffic Signal Control**
> > The reviewer correctly notes that our reinforcement fine-tuning procedure optimizes a one-step objective at the level of environment decisions. However, we respectfully clarify that this does not contradict the long-horizon nature of the traffic control problem itself.
> >
> > Traffic signal control is inherently a long-horizon sequential control problem, where queue lengths and congestion patterns evolve as a consequence of previous phase decisions. Even when the learning signal is computed locally at each decision step, the state representation already encodes accumulated system dynamics, including lane-level vehicle counts and queue states that reflect the effects of prior actions. Consequently, we train the policy on states that implicitly capture the long-term evolution of traffic.
> >
> > In other words, although we apply optimization updates at a single decision step, the training distribution of states arises from long-horizon traffic trajectories generated by the simulator. The policy therefore learns to map these long-horizon system states to appropriate actions. This training paradigm is common in several control settings where the environment itself captures temporal accumulation while we apply learning updates using individual decision steps.
> >
> > The use of one-step optimization for traffic signal control has been widely explored in recent studies and has become a commonly adopted setting. For example, \[1\] employs reward differencing with a one-step policy optimization scheme, Group Relative Turn-level Policy Optimization, while evaluating long-horizon travel-planning outcomes. Similarly, recent studies on large language model optimization in sequential decision settings have explored contextual-bandit style updates \[2\] even when the underlying task is multi-turn or sequential in nature. For instance, ShopSimulator \[3\] highlights the practical challenges of scaling from one-step optimization to stable multi-turn reinforcement learning in LLM environments, while other work \[2\] demonstrates that online contextual-bandit optimization can be effective even when the underlying task exhibits a multi-step structure, such as in iterative code generation settings. These studies suggest that locally optimized objectives can remain effective when the environment dynamics themselves encode temporal accumulation.
> >
> > The queue-difference reward shaping provides a locally informative signal that reflects the direction of congestion change rather than absolute congestion levels alone. It helps the policy distinguish whether a phase decision is improving or worsening traffic flow at that moment. The reward hurdle mechanism further suppresses weak or noisy reward fluctuations by filtering updates below a calibrated threshold. As a result, the policy is encouraged to reinforce actions that produce meaningful improvements in traffic flow while reducing the influence of marginal or unstable reward signals.
> >
> > Together, these mechanisms allow the training procedure to remain locally optimized while still capturing long-horizon traffic effects through the environment dynamics and state representation. Furthermore, the evaluation metrics used in our experiments reflect long-horizon traffic performance, including average queue length over an episode, total travel time, and overall throughput. Improvements in these metrics indicate that the learned policy produces better traffic dynamics over extended horizons, even though we apply training updates using information from the individual decision level. For these reasons, we believe that the contextual-bandit style optimization remains a reasonable and practical design choice for this setting.

---

> > > ### Author Response · Authors · 2026-03-04
> > > **Follow-up Response to Reviewer BDSA - Part 2**
> > >
> > > **Q2: Environment-Level Actor-Critic Learning**
> > > The reviewer raises a valid point that actor-critic approaches could, in principle, be applied at the environment level, for example, by learning a value function over traffic states and using advantage estimates to guide policy updates.
> > >
> > > We agree that such approaches are theoretically possible and represent an interesting direction for future work. Our intention was not to claim that environment-level actor-critic learning is fundamentally infeasible. Rather, our current design focuses on a simpler reinforcement fine-tuning formulation in which each decision exchange is treated independently during optimization.
> > >
> > > In OracleTSC, each control decision is generated through autoregressive token generation by the language model, after which the selected phase action receives environmental feedback. We compute updates during training at the token level for the generated response.
> > >
> > > Introducing an environment-level critic would require an additional learning component that estimates long-term value over traffic states and integrates these estimates into the token-level policy updates. Prior work, such as LLMLight \[4\], explores this direction by incorporating a value-function component alongside the language model; however, this approach relies on a pre-trained value model trained in conjunction with the LLM, which introduces additional supervision and architectural complexity. In contrast, we study whether LLM-based controllers can be trained directly through reinforcement fine-tuning without relying on external, pre-trained critic models.
> > >
> > > More broadly, multi-turn optimization frameworks such as WebGPT \[5\] and Trace \[6\] suggest promising directions for extending LLM-based optimization in sequential environments. For example, Trace demonstrates traffic optimization through simulation-based tuning of signal timing parameters. Exploring such multi-turn optimization strategies for traffic signal control is an interesting future direction. However, investigating these architectures would require substantial additional system design beyond the scope of the present work. Our focus here is therefore on establishing the feasibility of token-level reinforcement fine-tuning for LLM-based traffic signal control without external value models.
> > >
> > > Our goal in OracleTSC is to identify how to stabilize Reinforcement Fine-Tuning of LLM-based controllers under weak and noisy reward signals, a major practical challenge in prior work. The reward hurdle mechanism and semantic uncertainty regularization are introduced specifically to address this instability in the LLM training process.
> > >
> > > We therefore position our work as addressing stability and reward-signal quality in LLM-based control policies, rather than exploring the full design space of environment-level reinforcement learning architectures.
> > >
> > > **References**
> > >
> > > 1. Yuanzhe Shen, Zisu Huang, Zhengyuan Wang, Muzhao Tian, Zhengkang Guo, Chenyang Zhang, Shuaiyu Zhou, Zengjie Hu, Dailin Li, Jingwen Xu, Kaimin Wang, Wenhao Liu, Tianlong Li, Fengpeng Yue, Feng Hong, Cao Liu, and Ke Zeng. 2026\. TRIP-Bench: A Benchmark for Long-Horizon Interactive Agents in Real-World Scenarios. arXiv:2602.01675.
> > > 2. Ziru Chen, Dongdong Chen, Ruinan Jin, Yingbin Liang, Yujia Xie, and Huan Sun. 2026\. Bridging Online and Offline RL: Contextual Bandit Learning for Multi-Turn Code Generation. arXiv:2602.03806.
> > > 3. Pei Wang, Yanan Wu, Xiaoshuai Song, Weixun Wang, Gengru Chen, Zhongwen Li, Kezhong Yan, Ken Deng, Qi Liu, Shuaibing Zhao, Shaopan Xiong, Xuepeng Liu, Xuefeng Chen, Wanxi Deng, Wenbo Su, and Bo Zheng. 2026\. ShopSimulator: Evaluating and Exploring RL-Driven LLM Agent for Shopping Assistants. arXiv:2601.18225.
> > > 4. Siqi Lai, Zhao Xu, Weijia Zhang, Hao Liu, and Hui Xiong. 2024\. LLMLight: Large Language Models as Traffic Signal Control Agents. arXiv:2312.16044.
> > > 5. Reiichiro Nakano, Jacob Hilton, Suchir Balaji, Jeff Wu, Long Ouyang, Christina Kim,  Christopher Hesse, Shantanu Jain, Vineet Kosaraju, William Saunders, Xu Jiang, Karl Cobbe, Tyna Eloundou, Gretchen Krueger, Kevin Button, Matthew Knight, Benjamin Chess, and John Schulman. 2022.WebGPT: Browser-assisted question-answering with human feedback. arXiv:2112.09332.
> > > 6. Ching-An Cheng, Allen Nie, and Adith Swaminathan. 2024\. Trace is the Next AutoDiff: Generative Optimization with Rich Feedback, Execution Traces, and LLMs. arXiv:2406.16218.

---

### Review · Reviewer_CJng · 2026-02-04

**Summary Of Contributions:**

The paper focus on the problem of traffic signal control for optimal throughput. There is a long line of work on using RL methods to directly learn traffic control policies. More recently, some papers have used LLMs as traffic signal controllers, i.e. they output natural language text which is parsed and interpreted as making switching decisions. The present paper is related to this line of work. Given some open source base LLMs, they fine-tune (LoRA) an LLM using an RL signal. The technical contribution in the paper consists of some specialized reward shaping, as well as a special form of entropy regularization which measures confidence over decisions (by rolling out multiple trajectories) rather than over tokens. Experiments test the method and find that on the relevant performance metrics (delays, average queue length, travel time, and throughput) their new method consistently outperforms other language-based methods, and almost all prior RL methods.

**Audience:**

Yes

**Audience Explanation:**

This is an application of standard techniques to a rather niche problem, but some individuals in the audience would be interested to know about it.

**Broader Impact Concerns:**

There are no broader impact concerns.

**Claims And Evidence:**

Yes

**Claims Explanation:**

All methods are explained clearly. The experiments are thorough, including both ablation studies on the components of the method, tests with multiple different base LLMs, as well as reasonable comparisons to many prior baselines both traditional RL and natural-language based.

**Requested Changes:**

I have one small complaint. The method claims "interpretability" as an advantage of LLM/CoT based policies over traditional RL policies. This certainly makes intuitive sense: you can see the reasoning written in natural language. However, in other contexts, it has sometimes been the case that a reasoning LLM can generate a spurious CoT yet still consistently use this to get the right answer. CoT faithfulness is an open problem as I understand it. So how can we be completely sure the CoT here really is interpretable? I don't think this necessitates major changes to the paper (simply would strengthen the work in my view). Some brief discussion is probably warranted.

---

> ### Author Response · Authors · 2026-02-27
> **Response to Reviewer CJng**
>
> We thank the reviewer for the positive assessment of our contributions, clarity, and experimental coverage.
>
> **Interpretability and Faithfulness of Chain-of-Thought**
> We appreciate the reviewer’s thoughtful comment regarding interpretability. While language-based policies provide a more accessible interface for inspecting decisions than fully opaque neural controllers, we agree that chain-of-thought (CoT) explanations are not guaranteed to be mechanistically faithful. Ensuring that generated reasoning steps accurately reflect the model’s internal decision process remains an open research problem. We have added the following discussion in our revised paper.
>
> *The generated chain-of-thought is not guaranteed to be faithful to the model’s internal reasoning process. Ensuring mechanistic faithfulness in LLM reasoning remains an open research problem. Recent surveys of LLM reasoning frontiers highlight the complexity of scaling inference, supervising reasoning processes, and developing reliable agentic systems. Similarly, work on the extractive–abstractive spectrum demonstrates inherent trade-offs between abstraction, verifiability, and perceived utility in LLM generations. We interpret OracleTSC’s explanations as structured, human-readable rationales that improve transparency and behavioral consistency, rather than as formal guarantees of internal causal computation.*
>
> *Overall, the qualitative shift from hesitant exploration to structured and task-aligned rationales complements our quantitative improvements in traffic metrics, while acknowledging that reasoning faithfulness verification remains beyond the scope of the present work.*
>
> Recent work has examined both the benefits and limitations of CoT reasoning, including surveys of reasoning methods in large language models and approaches for verifying step-by-step reasoning via external checkers or process supervision \[4\]. Other studies highlight that abstract explanations can significantly increase perceived utility, even when not even when not fully supported by verifiable evidence or explicitly cited \[5\]. Methods such as verifier models, step-level reward modeling, and distillation from stronger reasoning systems have shown promise in improving reasoning reliability; however, a consensus solution for guaranteeing faithfulness has not yet emerged.
>
> In our experiments, we occasionally observe minor arithmetic inconsistencies (e.g., subtraction or aggregation errors) in reasoning traces produced by smaller models. These errors do not typically alter the selected phase decision, but they highlight the broader challenge of ensuring fully faithful reasoning in resource-constrained models.
>
> We therefore temper our interpretability claims: explanations generated by OracleTSC should be interpreted as structured, human-readable rationales rather than formal guarantees of internal causal computation. We have added a dedicated discussion in the qualitative results section clarifying this limitation.
>
> Developing robust verification mechanisms for step-by-step reasoning in control settings is an important direction for future work, but is beyond the scope of the current study.

---

### Review · Reviewer_qJzb · 2026-02-18

**Summary Of Contributions:**

This paper aims to develop an explainable and high-performance reinforcement learning (RL) method for traffic signal control (TSC).
Here, "explainable" refers to the use of a large language model (LLM) as an intermediary, leveraging natural language to interpret and guide decision-making.
While prior LLM-based TSC methods have struggled to make RL effective,
this paper addresses these challenges by introducing two key mechanisms: a reward hurdle mechanism and uncertainty regularization.
Through numerical experiments, the authors demonstrate that the proposed method, OracleTSC, achieves superior performance among explainable (i.e., LLM-based) TSC methods.

Key strengths:
- The experimental setup and underlying ideas are clearly described.
- The selection and adaptation of approaches to this task involve non-trivial engineering.

Key weaknesses:
- Source code for reproducing the experiments is not provided.
- The proposed method does not propose a novel algorithm but applies existing ideas to a specific task.
- While OracleTSC outperforms other explainable (LLM-based) methods, its performance remains inferior to non-explainable approaches.

**Audience:**

Yes

**Audience Explanation:**

This is somewhat borderline.
TSC as a task may be too domain-specific for the core TMLR audience, and the work might be a more natural fit for domain-specific or application-oriented venues (e.g., KDD).
From a machine learning perspective, the paper does not propose a novel algorithm in itself, but rather applies existing ideas to a specific task, which may be of limited interest to audience members primarily interested in theory or methodology.
On the other hand, the selection and adaptation of approaches to this task involve non-trivial engineering, and the motivation and specific methods are clearly explained.
Even for readers unfamiliar with TSC, those exploring new application domains for RL and LLM-based methods may find the findings of interest.

**Claims And Evidence:**

Yes

**Claims Explanation:**

The claims in this paper are purely empirical rather than theoretical; the main claim is that the proposed approach is effective for TSC as a specific application domain.
Ideally, the source code for reproducing the experiments should be provided as supporting evidence, but it is not available.
Nevertheless, the experimental setup and the underlying ideas are clearly described, and the results would be reasonably reproducible by experts in the field.

**Requested Changes:**

If possible, I would appreciate a response addressing the weaknesses mentioned above. This is not critical but would strengthen the paper.

Minor comments (non-critical):
- After Eq. (2): $(1-\epsilon, , 1+\epsilon)$ appears to contain an extra comma.
- In Eq. (10): an opening bracket $($ seems to be unclosed.
- In Eq. (11): $p$ appears in upright (roman) font rather than italic.

---

> ### Author Response · Authors · 2026-02-27
> **Response to Reviewer qJzb**
>
> We thank the reviewer for the thoughtful assessment and for highlighting both strengths and limitations.
>
> **Code Availability**
> We agree that releasing source code would further strengthen reproducibility. We are currently preparing a clean and well-documented version of the implementation for public release. The codebase will be made publicly available upon acceptance.
>
> **Novelty and Positioning**
> We acknowledge the reviewer’s point that OracleTSC applies existing ideas (baseline subtraction and entropy regularization) to a specific domain. In the revision, we carefully position the contribution as identifying and addressing a concrete failure mode of reinforcement fine-tuning in weak-reward, long-horizon control tasks, i.e., TSC,  rather than claiming algorithmic novelty in isolation. We partially agree with the reviewer’s feedback regarding the algorithm’s design, which we carefully discuss in our revised manuscript under Related Work:
>
> *While PPO and reward shaping have been widely studied in reinforcement learning, their direct application to LLM-based traffic signal control remains unstable under weak and delayed reward signals. Our work does not introduce a fundamentally new RL algorithm, but rather demonstrates how targeted reward thresholding and uncertainty control can address these stability failures in this specific regime.*
>
> We interpret the reward hurdle as an explicit and calibrated form of baseline subtraction tailored to regimes where most actions produce marginal reward differences. In such settings, standard PPO stabilization techniques (e.g., clipping, advantage normalization) are insufficient to prevent early suboptimal actions from dominating learning dynamics. By reframing baseline selection as a first-class design component, the hurdle mechanism systematically suppresses suboptimal updates and improves training stability in long-horizon TSC.
>
> In parallel, we distinguish between token-level entropy regularization and semantic-level (phase-level) uncertainty control. Rather than applying generalized token entropy, we introduce discrete, temperature-scaled semantic entropy at the action level, which better aligns uncertainty regularization with the structure of the control problem. This provides a clearer coupling between decision confidence and environmental reward.
>
> Thus, while the building blocks are rooted in classical methods, our work contributes new insights into how baseline subtraction and entropy regularization should be jointly structured for stable reinforcement fine-tuning of LLM controllers in weak-feedback environments.
>
> **Performance Relative to Non-Explainable Methods**
> We agree that fully optimized, non-explainable RL methods achieve stronger performance on certain benchmarks, and our results do not surpass the best classical controllers in absolute efficiency. Our objective is not to claim superiority over all RL approaches, but to stabilize and improve the autonomy of LLM-based traffic signal controllers. Prior pioneering work applying LLMs to TSC \[6,7\] often relies on external critic models, trajectory filtering, or hybrid integration with conventional TSC algorithms to compensate for instability in reinforcement fine-tuning. Such dependencies limit autonomy and scalability and arise because long-term utility in TSC becomes apparent only after many timesteps, making stable long-horizon behavior difficult to elicit through naive PPO or gold-standard CoT supervision. Our primary contribution is to directly stabilize LLM reinforcement training in this weak-feedback regime through the introduction of the reward hurdle mechanism, reducing policy collapse without requiring external scaffolding. We revise the discussion in our Related Work to more explicitly frame OracleTSC as improving the competitiveness and self-sufficiency of explainable LLM-based controllers, rather than claiming superiority over all RL methods.
>
> *Importantly, our objective is not to claim superiority over all existing RL algorithms or fully optimized black-box RL systems. Highly optimized, non-explainable RL controllers remain strong on several benchmarks. Rather, our goal is to improve the performance, stability, and reliability of explainable LLM-based controllers while preserving natural-language reasoning and semantic-level uncertainty modeling.*
>
> **Minor Issues**
>
> - We remove the extra comma after the definition of the trust region in Equation 2\.
> - We added an extra ‘)’ to close the unclosed bracket in Equation 10 of the optimal constant baseline.
> - We italicize ‘p’ in the definition of Temperature-scaled Softmax DSE in Equation 11 to match the rest of the text.

---

### Review · Reviewer_6cyR · 2026-02-18

**Summary Of Contributions:**

This paper introduces OracleTSC, a framework for fine-tuning LLMs for traffic signal control using reinforcement learning. The authors identify two key challenges in applying PPO to LLM-based TSC: (1) weak reward signals where most actions produce marginal improvements, and (2) reasoning drift where uncertain models generate inconsistent phase selections across responses. To address these issues, OracleTSC proposes a Reward Hurdle Mechanism (RHM) to subtract a calibrated threshold from environmental rewards to suppress low-impact actions and temperature-scaled softmax discrete semantic entropy for measuring uncertainty at the semantic action level (traffic phase) rather than token level. Experiments on LibSignal benchmark with Qwen3 and LLaMA3-8B show substantial improvements, a significant decrease in queue lengths, with strong cross-intersection generalization.

**Additional Comments:**

In addition to non-explainable RL and LLM-based decision-making, recent progress has been made in algorithm design using LLMs. The resulting solution, typically presented as code and/or thought, offers potential benefits in both explainability and real-time responsiveness.

A systematic survey on large language models for algorithm design. ACM Computing Surveys 2026
Evolution of Heuristics: Towards Efficient Automatic Algorithm Design Using Large Language Model. ICML 2024.

**Audience:**

Yes

**Audience Explanation:**

The two mechanisms can be useful to other RL tasks that have weak rewards and reasoning drift.

**Claims And Evidence:**

Yes

**Claims Explanation:**

The authors have compared with both RL and LLM-based baselines on benchmark instances. They also conducted an ablation study to validate the contribution of different components.

**Requested Changes:**

The hurdle rate requires empirical tuning based on reward distribution analysis. Table 4 shows different values across models/intersections (2.7-3.7). While the authors acknowledge this limitation, discussing potential adaptive mechanisms (e.g., based on recent reward percentiles) would strengthen the paper.

Is there a trade-off between performance and computational cost as the number of response samples G varies? Discuss the computational-accuracy trade-off for different numbers of response samples.

Could you provide a detailed illustration of one instance, showing the inputs, decision-making process, and output?

---

> ### Author Response · Authors · 2026-02-27
> **Response to Reviewer 6cyR - Part 1**
>
> We thank the reviewer for the positive assessment of our contributions and ablations, as well as for the constructive suggestions on other directions to explore.
>
> **Adaptive Hurdle Mechanisms**
> We agree that the hurdle rate requires empirical tuning and that adaptive calibration is a natural extension. To directly evaluate the reviewer’s suggestion, we conducted an additional experiment in which the hurdle rate was dynamically updated at the end of each episode using the 70th percentile of the queue-difference rewards stored in the replay buffer.
>
> Table 7: Effect of dynamically adjusting the hurdle using the 70th percentile of queue-difference rewards (Reward \= Queue Difference \- Hurdle \+ Temperature-scaled Softmax DSE, LLaMA3-8B). Evaluated on CityFlow 1x1.
>
> |  | Travel Time | Queue Length | Delay | Throughput |
> | :---- | ----- | ----- | ----- | ----- |
> | Baseline | 594.9 | 138.63 | 0.81 | 1359 |
> | Episode 1 | **431.9** | **119.97** | 0.81 | **1591** |
> | Episode 2 | 568.4 | 138.14 | **0.68** | 1412 |
> | Episode 3 | 1258.8 | 191.63 | 0.75 | 577 |
> | Episode 4 | 1339.2 | 196.12 | 0.73 | 726 |
>
> From Table 7, while the first episode achieves moderate performance, subsequent episodes exhibit clear instability, with travel time and queue length degrading significantly by Episode 3\. The adaptive percentile-based hurdle introduces large shifts in the effective reward baseline across episodes, which amplifies training variance rather than stabilizing it in this setting.
>
> These results suggest that naive percentile-based adaptive calibration does not automatically improve stability and may, in fact, destabilize learning in weak-reward regimes. In contrast, a carefully calibrated fixed hurdle provides more consistent gradient filtering and improved convergence. We include these results in the supplementary appendix for completeness.
>
> In addition to this, we also evaluated an adaptive impact for the KL divergence weight. As a partial step toward adaptive control, we evaluated dynamic adjustment of the KL penalty weight during training (Table 8), which modulates the relative strength of the environmental reward signal and therefore influences the effective variance of policy-gradient updates. Since baseline selection directly impacts gradient variance \[1\], this provides a practical mechanism for controlling training stability alongside a fixed hurdle rate.
>
> Table 8: Effect of dynamic KL penalty weight (β) (Reward \= Queue Difference \- Hurdle Rate \= 2.5, Qwen3-0.6B). Evaluated on CityFlow 1x1.
>
> |  | Travel Time | Queue Length | Delay | Throughput |
> | :---- | ----- | ----- | ----- | ----- |
> | Baseline | 506.1 | 104.06 | 0.71 | 1485 |
> | Episode 1 (β=0.05) | 249 | 74.44 | 0.71 | 1813 |
> | Episode 2 (β=0.025) | **137** | **39.95** | **0.67** | **1961** |
>
> **Trade-off with Number of Responses**
> Regarding the computational–accuracy trade-off, increasing the number of sampled responses G substantially raises GPU memory consumption and training time. In the revision, we clarify this trade-off and expand the discussion of practical choices for G. Specifically, we include additional experiments with G=2 and G=4, and compare them against the original setting of G=8, to better characterize the performance-efficiency balance.
>
> Table 9: Effect of Number of Responses (G) (Reward \= Queue Difference \- Hurdle \+ Temperature-scaled Softmax DSE, LLaMA3-8B). Evaluated on CityFlow 1x1.
>
> | Number of Responses (G) | Travel Time | Queue Length | Delay | Throughput |
> | :---- | ----- | ----- | ----- | ----- |
> | 2 | 377 | 82.04 | 0.73 | 1665 |
> | 4 | 315.5 | 103.86 | 0.76 | 1750 |
> | 8 | **146.9** | **45.5** | **0.67** | **1946** |
>
> As G increases, performance improves consistently in terms of travel time and throughput. The G=8 configuration achieves the strongest results, reducing travel time by approximately 41% compared to G=2. However, this comes at significantly higher computational cost, as memory usage scales approximately linearly with G.
>
> These results indicate a clear performance-efficiency trade-off: smaller G values reduce computational burden but weaken semantic consensus regularization, while larger G values improve stability and final policy quality at increased cost. We include this expanded discussion in the revision to clarify practical deployment considerations.
>
> **Detailed Illustration of Instance**
> We appreciate the request for a detailed example. In the revision, we add a step-by-step illustration of a representative decision, including the traffic state input, the model’s reasoning process, and the selected phase action in Figure 2\.
>
> Finally, we thank the reviewer for highlighting recent progress in LLM-based algorithm design. We agree that these directions are highly relevant and incorporate the suggested references \[8,9\] and broader perspective into the related work and discussion as follows:

---

> > ### Author Response · Authors · 2026-02-27
> > **Response to Reviewer 6cyR - Part 2**
> >
> > *“Beyond decision-making and control, recent work has explored using LLMs for automatic algorithm design and heuristic discovery. A recent systematic survey highlights how LLMs can generate executable algorithms, code-based solutions, and structured reasoning processes that improve both interpretability and responsiveness. Similarly, demonstrates that LLMs can iteratively refine algorithmic strategies, enabling efficient automatic design of problem-specific solvers. These approaches focus primarily on synthesizing or evolving discrete algorithmic procedures. In contrast, traffic signal control requires stable policy optimization under weak, delayed reward feedback in a stochastic control environment. Our work is complementary: rather than designing new symbolic algorithms, we study how to directly stabilize reinforcement fine-tuning of LLM-based controllers, enabling autonomous, interpretable decision-making without reliance on external critics or hybrid ensembles.”*

---

### Author Response · Authors · 2026-02-27
**OracleTSC: Response to Reviewers**

Journal: Transactions on Machine Learning Research
Paper: [https://openreview.net/forum?id=WmJu5MkoQD](https://openreview.net/forum?id=WmJu5MkoQD)
Title: OracleTSC: Oracle-Informed Reward Hurdle and Uncertainty Regularization for Traffic Signal Control

**To Reviewers and Editors:**

We sincerely thank the Action Editor and the Reviewers for their careful reading of our manuscript and for the constructive feedback. We found the comments highly valuable, and they have helped us identify several areas where the presentation and empirical justification of OracleTSC can be strengthened.

Overall, the reviewers raised four central themes:

1) Interpretation of LLM-based explanations and the limits of chain-of-thought (CoT) faithfulness,
2) Hyperparameter sensitivity and how to set key parameters adaptively,
3) Motivation, necessity, and positioning of the proposed reward hurdle mechanism relative to standard PPO stabilization, and
4) Experimental rigor and the interpretation of queue-difference rewards under varying traffic demand patterns.

In the revised manuscript, we address these concerns directly through clearer framing, additional experimental evidence, and a more comprehensive discussion of methodological limitations. Due to the 5000-character limit per response, we have organized our replies into multiple sections.

We thank the Action Editor and Reviewers again for the detailed critique. We believe the planned revisions (tempered interpretability claims, stronger justification, clearer positioning of the hurdle mechanism, and clearer framing of queue-difference shaping under demand drift) have substantially improved the rigor and impact of the manuscript.

**References**

1. Lex Weaver and Nigel Tao. 2001\. The optimal reward baseline for gradient-based reinforcement learning. In Proceedings of the Seventeenth conference on Uncertainty in artificial intelligence (UAI'01). Morgan Kaufmann Publishers Inc., San Francisco, CA, USA, 538-545.
2. Evan Greensmith, Peter L. Bartlett, and Jonathan Baxter. 2004\. Variance reduction techniques for gradient estimates in reinforcement learning. J. Mach. Learn. Res., 5:1471-1530.
3. Trung Quoc Luong, Xinbo Zhang, Zhanming Jie, Peng Sun, Xiaoran Jin, Hang Li. 2024\. ReFT: Reasoning with Reinforced Fine-Tuning. arXiv:2401.08967.
4. Zixuan Ke, Fangkai Jiao, Yifei Ming, Xuan-Phi Nguyen, Austin Xu, Do Xuan Long, Minzhi Li, Chengwei Qin, Peifeng Wang, Silvio Savarese, Caiming Xiong, and Shafiq Joty. 2025\. A Survey of Frontiers in LLM Reasoning: Inference Scaling, Learning to Reason, and Agentic Systems. arXiv:2504.09037.
5. Theodora Worledge, Tatsunori Hashimoto, and Carlos Guestrin. 2024\. The Extractive-Abstractive Spectrum: Uncovering Verifiability Trade-offs in LLM Generations. arXiv:2411.17375.
6. Siqi Lai, Zhao Xu, Weijia Zhang, and Hao Liu and Hui Xiong. 2024\. LLMLight: Large Language Models as Traffic Signal Control Agents. arXiv:2312.16044.
7. Maonan Wang, Aoyu Pang, Yuheng Kan, Man-On Pun, Chung Shue Chen, and Bo Huang. 2024\. Llm-assisted light: Leveraging large language model capabilities for human-mimetic traffic signal control in complex urban environments. arXiv:2403.08337.
8. Fei Liu, Yiming Yao, Ping  Guo, Zhiyuan Yang, Xi Lin, Zhe Zhao, Xialiang Tong, Kun Mao, Zhichao Lu, Zhenkun Wang, Mingxuan Yuan, and Qingfu Zhang. 2026\. A Systematic Survey on Large Language Models for Algorithm Design. ACM Comput. Surv. 218\. [https://doi.org/10.1145/3787585](https://doi.org/10.1145/3787585).
9. Fei Liu, Xialiang Tong, Mingxuan Yuan, Xi Lin, Fu Luo, Zhenkun Wang, Zhichao Lu, and Qingfu Zhang. 2024\. Evolution of heuristics: towards efficient automatic algorithm design using large language model. Proceedings of the 41st International Conference on Machine Learning. 1304\.

---

> ### Comment · Action_Editor_Wb3x · 2026-03-21
> **Some concerns still persist**
>
> Dear Authors,
>
> Thanks for the rebuttal. I was going over the reviewers' concerns. However, some of the concerns have not been addressed.
>
> 1. The authors have acknowledged (in response to reviewer BDSA) that the hurdle rate is similar to the variance reduction techniques typically used for policy gradient-based algorithms (like subtracting the value function as a baseline). Given that, the paper should conduct a relative study of the necessity of this compared to the other existing techniques. At least, the paper should not claim this as one of the key contributions unless they convince the readers of the novelty of this.
>
> 2. The authors should provide results over multiple seeds; taking too long a time is not an excuse for a journal paper. Or at least show that, at least on a small scale, it is not an issue. What many studies are doing does not mean that they are the correct approach.
>
> 3. There are some discussions about the contextual bandit, and the long term horizon <https://openreview.net/forum?id=WmJu5MkoQD&noteId=1JTLQTwcJs> Can the authors clarify that? Did the authors answer these concerns in the main file?
>
>
> AE

---

> > ### Author Response · Authors · 2026-03-21
> > **We will respond shortly.**
> >
> > Dear AE,
> >
> > Thank you for providing the feedback.
> >
> > We aim to get back to you shortly during the weekend.
> >
> > Best,
> >
> > Authors

---

> > ### Author Response · Authors · 2026-03-21
> > **Follow-up Response to Action Editor**
> >
> > We thank the Action Editor for the careful reading and for highlighting the remaining concerns. We address and clarify each point below and have revised the manuscript accordingly. For several follow-up points raised after our initial revision, we deferred additional changes until we received further guidance from the reviewers and the Action Editor. Based on your comments, we have now incorporated the corresponding revisions.
> >
> > **1\. Positioning of the Reward Hurdle Mechanism**
> > We agree with the Action Editor that the reward hurdle mechanism is closely related to classical variance reduction techniques in policy gradient methods, particularly constant baseline subtraction. In the revised manuscript, we have adjusted our claims to reflect this more accurately. Specifically, we no longer present the hurdle as a standalone novel algorithmic contribution. Instead, we position it as an explicit and calibrated form of baseline subtraction, and as an empirically motivated design choice that improves stability in long-horizon LLM-based control (Section 2). This connection is discussed in the manuscript (Section 3.3, Connection to Variance Reduction Techniques).
> >
> > To address the concern regarding necessity, we have strengthened the empirical justification through ablations included in the revised paper (Tables 4-7), which show:
> >
> > - Standard PPO (with extensive hyperparameter tuning over learning rate, GAE-$\\lambda$, and reward discount factors) and REINFORCE do not consistently improve performance,
> > - Our reward hurdle mechanism and uncertainty minimization rewards significantly improves stability and downstream metrics (travel time, queue length, throughput).
> >
> > We note that, consistent with TMLR’s acceptance criteria, our revisions focus on ensuring that claims are supported by clear empirical evidence rather than emphasizing novelty in isolation. Accordingly, we have adjusted the presentation to highlight the empirical insight and practical value of the approach.
> >
> > **Multi-Seed Evaluation**
> > We agree that multi-seed evaluation is important for RL. In response, we have added multi-seed results (3 seeds) for the primary configuration (LLaMA3-8B, CityFlow1x1), now included in the revision (Table 11). These results show consistent performance improvements across seeds, indicating that gains are not due to a single favorable initialization. We acknowledge that larger-scale evaluation would further strengthen statistical confidence and explicitly note this as a limitation in the revised manuscript (Section A.8). Our goal here is to provide initial evidence of robustness at a small scale, as suggested.
> >
> > **Contextual Bandit vs. Long-Horizon Control**
> > We thank the Action Editor for pointing out the need for clarification. This concern has now been explicitly addressed in the main paper. In Section 3.2, we revised the training objective to clearly state that training is performed at the level of individual state-response exchanges:
> >
> > *Training proceeds at the level of individual state-response exchanges. Given a traffic state $s\_t$, the LLM generates a single response trajectory $\\kappa \= {(s\_0, a\_0), . . . ,(s\_O, a\_O)}$ consisting of reasoning tokens followed by a phase selection. Each exchange is treated as an independent training instance, similar in spirit to contextual bandit-style reinforcement fine-tuning used in RLHF. We do not propagate gradients across multiple environment steps or perform trajectory-level temporal credit assignment over extended control horizons*
> >
> > This formulation does not contradict the long-horizon nature of traffic signal control, since:
> >
> > - The environment dynamics are inherently sequential, and each state encodes accumulated effects of prior decisions,
> > - Training data is drawn from long-horizon trajectories generated by the simulator, and
> > - Evaluation metrics (travel time, delay, throughput) reflect long-horizon performance.
> >
> > Additionally, recent work shows that one-step or contextual-bandit optimization can remain effective in sequential settings, particularly when environment dynamics encode temporal accumulation \[1-3\].

---

> > > ### Author Response · Authors · 2026-03-21
> > > **Follow-up Response to Action Editor - 2**
> > >
> > > **Summary**
> > > In response to the Reviewer's and Action Editor’s feedback, we have:
> > >
> > > - Tempered claims regarding the novelty of the hurdle mechanism and clarified its relationship to baseline subtraction,
> > > - Added multi-seed results to improve experimental rigor, and
> > > - Explicitly clarified the training formulation (contextual bandit vs. long-horizon control) in the main manuscript.
> > >
> > > We believe these revisions address the remaining concerns and better align the paper with TMLR’s criteria of clear, evidence-supported claims. We thank the Action Editor again for the guidance.
> > >
> > > **References**
> > >
> > > 1. Yuanzhe Shen, Zisu Huang, Zhengyuan Wang, Muzhao Tian, Zhengkang Guo, Chenyang Zhang, Shuaiyu Zhou, Zengjie Hu, Dailin Li, Jingwen Xu, Kaimin Wang, Wenhao Liu, Tianlong Li, Fengpeng Yue, Feng Hong, Cao Liu, and Ke Zeng. 2026\. TRIP-Bench: A Benchmark for Long-Horizon Interactive Agents in Real-World Scenarios. arXiv:2602.01675.
> > > 2. Ziru Chen, Dongdong Chen, Ruinan Jin, Yingbin Liang, Yujia Xie, and Huan Sun. 2026\. Bridging Online and Offline RL: Contextual Bandit Learning for Multi-Turn Code Generation. arXiv:2602.03806.
> > > 3. Pei Wang, Yanan Wu, Xiaoshuai Song, Weixun Wang, Gengru Chen, Zhongwen Li, Kezhong Yan, Ken Deng, Qi Liu, Shuaibing Zhao, Shaopan Xiong, Xuepeng Liu, Xuefeng Chen, Wanxi Deng, Wenbo Su, and Bo Zheng. 2026\. ShopSimulator: Evaluating and Exploring RL-Driven LLM Agent for Shopping Assistants. arXiv:2601.18225.

---

> > > ### Comment · Action_Editor_Wb3x · 2026-03-24
> > > **Thanks for the clarification**
> > >
> > > Dear Authors,
> > >
> > > Thank you for your responses. Regarding the multi-seed evaluation, I saw that the result indeed changed quite a bit (Table 11). Can you please perform more extensive evaluation to study the effect? Thanks,
> > >
> > > AE

---

> ### Author Response · Authors · 2026-04-02
> **Follow-up Response to Action Editor - 3**
>
> We thank the Action Editor for the continued feedback regarding variance and experimental robustness. In response, we conducted additional multi-seed experiments to further evaluate the stability of our approach. The results across independent seeds are summarized below.
>
> |  | Travel Time | Queue Length | Delay | Throughput |
> | :---- | :---- | :---- | :---- | :---- |
> | Seed=0 | 146.9 | 45.5 | 0.67 | 1946 |
> | Seed=11 | 173.2 | 53.65 | 0.66 | 1975 |
> | Seed=120 | 171.4 | 77.35 | 0.66 | 1851 |
> | Seed=2368 | 157.4 | 46.89 | 0.68 | 1936 |
> | Seed=6446 | 243.5 | 82.65 | 0.66 | 1844 |
> | Seed=8125 | 199.8 | 69.73 | 0.68 | 1900 |
> | Mean | 182.03 | 62.63 | 0.67 | 1908.67 |
> | Std. Dev. | 35.00 | 16.06 | 0.01 | 53.15 |
>
> Each full training run requires approximately one day per training episode on a single GPU, and each run consists of four training episodes. Given limited GPU availability, conducting large-scale multi-seed experiments is computationally expensive. We therefore conducted these additional seeds within the available revision timeline to provide further evidence of robustness.
>
> Across seeds, we observe variability in queue length, which is expected in reinforcement learning settings. However, the key performance metrics remain consistently strong: travel time remains substantially improved compared to the pretrained baseline across all seeds, throughput remains high and stable (1844-1975), and delay remains tightly controlled (0.66-0.68).
>
> Furthermore, performance remains competitive with, and in several cases improves upon, strong baselines including specialized methods such as MaxPressure and non-explainable RL approaches (e.g., MAPG, IPPO), as reported in Table 1 of the main paper. These results indicate that, despite inherent variance, the proposed method yields robust and consistent improvements in core traffic efficiency metrics across multiple random initializations.

---

> > ### Comment · Action_Editor_Wb3x · 2026-04-07
> > **Thanks for your effort**
> >
> > I do not have any more comments.

---

> > > ### Author Response · Authors · 2026-04-07
> > > **Thank you for the feedback**
> > >
> > > Dear AE,
> > >
> > > Thank you for the update.
> > >
> > > We look forward to the future steps.
> > >
> > > Best,
> > > Authors

---

### Decision · Action_Editor_Wb3x · 2026-04-07

**Recommendation:** Accept as is

**Audience:**

Yes

**Audience Explanation:**

This will be of interest to the system practitioners and traffic signal controllers.

**Claims And Evidence:**

Yes

**Claims Explanation:**

OracleTSC is a reinforcement fine-tuning framework for LLM-based traffic signal control. The paper starts from a concrete empirical observation: standard PPO is unstable in this setting because traffic control produces weak, delayed, and noisy rewards, and LLM controllers can also suffer from reasoning drift, where multiple sampled responses for the same traffic state lead to inconsistent phase choices. To address those two failure modes, the paper proposes two simple mechanisms: a Reward Hurdle Mechanism (RHM) that subtracts a calibrated threshold from the environmental reward so that only sufficiently impactful actions receive positive reinforcement, and a temperature-scaled semantic-entropy regularizer that penalizes uncertain or inconsistent answer modes across sampled responses. These are combined inside PPO to produce a more stable and consistent LLM controller while still preserving natural-language explanations.

Strengths:

The strongest reason is that the paper has a clear mechanism-to-result story. The ablation table shows that raw environmental reward alone gives only limited gains, while the hurdle term produces a much larger jump, and the best results come from combining the hurdle with uncertainty regularization. The paper reports that on CityFlow-1×1, fine-tuned LLaMA3-8B reduces average travel time by about 75% and queue length by about 67% relative to the pretrained baseline, while remaining competitive with strong specialized controllers.

The paper does not stop at one benchmark intersection; it evaluates zero-shot transfer across distinct topologies. A policy trained on Cologne1 improves CityFlow1×1 travel time from 594.9 to 517.5 and throughput from 1359 to 1427, while a policy trained on CityFlow1×1 improves Cologne1 travel time from 74 to 61.3 and queue length from 13.52 to 8.27 without additional finetuning. That is important because it suggests the model is learning something more transferable than intersection-specific memorization.


Regarding the Reviewers' concerns:
 Reviewers have raised some concerns. For example, the paper only considers a stage-based decision rather than a sequential long-term horizon-based decision, even though that is their original motivation. There are also some concerns regarding experimentation on multiple seeds. The authors have clarified that. I would suggest to add those discussion in the main version.

---

> ### Author Response · Authors · 2026-04-12
> **thank you & thank you**
>
> Dear AE and Reviewers,
>
> Thank you very much for your detailed and insightful comments.
>
> We have greatly benefited from your feedback and the review process.
>
> We will carefully incorporate your suggestions and finalize the submission in the coming days.
>
> It has been a truly valuable and enjoyable experience working with all of you throughout this process.
>
> Best regards,
> The Authors